# Ridge Boosting is Both Robust and Efficient

**David Bruns-Smith**
(Stanford University)

**Zhongming Xie**
(UC Berkeley)

**Avi Feller**
(UC Berkeley)

## Abstract

Estimators in statistics and machine learning must typically trade off between efficiency, having low variance for a fixed target, and distributional robustness, such as *multiaccuracy*, or having low bias over a range of possible targets. In this paper, we consider a simple estimator, *ridge boosting*: starting with any initial predictor, perform a single boosting step with (kernel) ridge regression. Surprisingly, we show that ridge boosting simultaneously achieves both efficiency and distributional robustness: for target distribution shifts that lie within an RKHS unit ball, this estimator maintains low bias across all such shifts and has variance at the semiparametric efficiency bound for each target. In addition to bridging otherwise distinct research areas, this result has immediate practical value. Since ridge boosting uses only data from the source distribution, researchers can train a single model to obtain both robust and efficient estimates for multiple target estimands at the same time, eliminating the need to fit separate semiparametric efficient estimators for each target. We assess this approach through simulations and an application estimating the age profile of retirement income.

## 1 Introduction

Estimators in statistics and machine learning must typically trade off between efficiency and robustness. Efficient estimators, largely developed in semiparametric statistics and econometrics, focus on having the smallest asymptotic variance (the "efficient variance") among unbiased estimates for a single target estimand. Importantly, such estimators provide valid asymptotically Normal confidence intervals—critical in many empirical applications—and these intervals have the smallest possible width. By contrast, robust estimators, the focus of an active literature in Distributionally Robust Optimization (DRO) and other subfields, instead aim to have good performance for many, possibly unspecified targets. For example, Kim et al. [2022] show that a class of "multi-accurate" estimators—based on boosting an initial predictor—constrains worst-case bias for predicting the unknown mean in a large class of covariate shift problems. In general, we expect that controlling worst-case bias across many estimands would come at the cost of increased variance.

Surprisingly, we show that a simple version of boosting, *once-boosting with ridge regression*, is simultaneously robust over a large set of possible distribution shifts, while also achieving the efficient variance and smallest possible confidence interval for each estimation target separately. Constructing this ridge boosting predictor is simple: we start with any initial predictor, and then perform one step of boosting using ridge regression in a Reproducing Kernel Hilbert Space (RKHS). For all target populations whose density ratio with respect to the source population is well-approximated by the RKHS, the resulting estimator is approximately unbiased and achieves the semiparametric efficiency bound. This is a very general (but not completely general) class of distribution shifts: it includes any shift whose density ratio can be expressed as linear in a fixed transformation of the covariates, even infinite-dimensional transformations. For example, this includes distribution shifts whose density ratio can be approximated as linear in the last-layer embedding of a pre-trained large language model, but not the more general class that would involve fine-tuning the neural network.

39th Conference on Neural Information Processing Systems (NeurIPS 2025).

We similarly establish this result for a broad class of linear estimands, generalizing the results from Kim et al. [2022] beyond the missing mean to more complex targets like average derivatives and impulse responses. In this more general setting, we replace the density ratio with the more flexible *Riesz representer* corresponding to the estimand of interest [Chernozhukov et al., 2021]. Our key technical insight is that kernel ridge regression implicitly estimates the Riesz representer needed for semiparametric estimation: the ridge boosting estimator we analyze is in fact numerically equivalent to the "Automatic Debiased Machine Learning" estimator of Chernozhukov et al. [2021] and inherits its optimality properties. As a result, we can train a single predictor using only source distribution data. Deploying this predictor to estimate any target estimand (whose Riesz representer is in the RKHS) will then have both low bias under distribution shift and asymptotically optimal confidence intervals—without ever explicitly computing target-specific bias correction terms.

Our results have immediate practical implications. In settings where practitioners must estimate many related quantities under different covariate shifts—such as estimating health outcomes across multiple hospitals, or computing age profiles of economic variables—our approach eliminates the need to fit separate semiparametric efficient estimators for each target. As we show in simulations, this approach also yields valid confidence intervals for scalar estimands, an important requirement for many applications.

**Paper organization.** The paper proceeds as follows. Section 2 formalizes the problem setup, defining our estimation targets and contrasting robustness and efficiency. Section 3 introduces ridge boosting and proves it is both multiaccurate and semiparametrically efficient. Section 4 demonstrates performance through simulations and an empirical application. Section 5 concludes with limitations and future directions.

## 1.1 Related literature

**Multiaccuracy and Multicalibration:** Multi-calibration, introduced by Hébert-Johnson et al. [2018], is a refinement of group calibration that requires a predictor to be simultaneously calibrated across a rich collection of (potentially overlapping) subpopulations. For a prediction task, calibration requires that among the individuals which receive prediction $f(x) = v$, the true expectation is $v$. Variants of the original definition have been studied by a number of works [Kim et al., 2019, 2022, Deng et al., 2023, Jung et al., 2021, Gopalan et al., 2022]). Multiaccuracy [Kim et al., 2019] is a weaker version of multi-calibration: it weakens multi-calibration by removing conditioning on the predicted values. Both concepts strengthen classical group fairness by ensuring fine-grained predictive accuracy without sacrificing overall performance. The multiaccuracy criterion is a special case of DRO [Hastings et al., 2024]. The link between multicalibration and boosting is discussed extensively in Globus-Harris et al. [2023]. Long et al. [2025] consider boosting over an RKHS to achieve multiaccuracy, but for classification. It would be interesting to see whether or not we could extend our result to their setting.

**Semiparametric efficiency and doubly robustness:** Semiparametric efficiency theory provides a rigorous foundation for the efficient estimation of target parameters in models that incorporate both parametric and nonparametric components [Bickel et al., 1993, Newey, 1994]. In the context of causal inference, doubly robust estimators [Robins et al., 1994, Kennedy, 2024] form a central class of methods that can attain semiparametric efficiency under correct specification of both. One motivation for these estimators comes from orthogonal (or Neyman-orthogonal) estimating equations [Chernozhukov et al., 2018, 2021, Foster and Syrgkanis, 2023], which reduce sensitivity to errors in nuisance function estimation. A complementary line of work [Zubizarreta, 2015, Ben-Michael et al., 2021, Athey et al., 2018, Hirshberg and Wager, 2021, Bruns-Smith et al., 2025a] focuses on balancing weights, which aim to reweight samples so that covariate distributions are matched across treatment groups. When balancing weights are combined with outcome regression, the resulting augmented estimators inherit both double robustness and semiparametric efficiency. In parallel, targeted maximum likelihood estimation (TMLE) [Van Der Laan and Rubin, 2006, Van der Laan et al., 2011] shows that, by incorporating a targeting step based on the efficient influence function of the parameter of interest and grounded in likelihood theory, TMLE achieves semiparametric efficiency. Cho et al. [2024] consider the TMLE update in an RKHS, and find a closely related universal adaptability property. In future work, it may be possible to unify their results with our boosting and multicalibration setting.

**Connection between multicalibration and causal inference:** There are several recent papers discussing the connection between multicalibration and causal inference. Wu et al. [2024] show the connection between invariant risk minimization and multicalibration in the context of concept shift. Ye and Li [2024] explores multicalibration and universal adaptability in survival analysis. Kern et al. [2024] shows that the multi-accurate conditional average treatment effect estimate is robust to unknown covariate shifts. Van Der Laan et al. [2023] also calibrate a baseline model to achieve semiparametric efficiency, albeit without using multicalibration.

## 2    Problem setup: Robustness vs efficiency

### 2.1    Notation

Let $X \in \mathcal{X}$ denote covariates and $Y \in \mathcal{Y} \subseteq \mathbb{R}$ an outcome of interest. We consider a source distribution $P$ over $(X, Y)$. We assume that we have $n_p$ independent and identically distributed observations from $P$, denoted by $\{(X_i, Y_i)\}_{i=1}^{n_p} \sim_{\text{i.i.d.}} P$. We let $X_p \in \mathbb{R}^{n_p \times d}$ denote the matrix of observed covariates and $Y_p \in \mathbb{R}^{n_p}$ the corresponding vector of outcomes. Define $\gamma_0(x) := \mathbb{E}_P[Y|X=x]$, the optimal mean-squared error predictor of $Y$ given $X$ in $P$.

### 2.2    Defining our estimation target

We consider the goal of estimating a scalar summary of the optimal predictor $\gamma_0$ [see Chernozhukov et al., 2018]. Examples include estimating a missing mean under covariate shift, estimating an average treatment effect, and estimating an average derivative. This setup generalizes the estimands considered in Kim et al. [2022].

**Definition 1** (Target Estimand). *For any function $f : \mathcal{X} \to \mathcal{Y}$, define:*

$$\theta_{target}(f) := \mathbb{E}_P[m(f, X)],$$

*where $m$ is some real-valued function of $f$ and $X$ such that $\theta_{target}$ is linear in $f$. Our target estimand is:*

$$\theta_0 := \theta_{target}(\gamma_0).$$

**Assumption 1** (Continuity). *We assume that $\theta_{target}$ is a continuous linear functional. That is, there exists a constant $C > 0$ such that:*

$$\theta(f)^2 \leq C\mathbb{E}_P[f(X)^2],$$

*for all $f$ with $\mathbb{E}[f(X)^2] < \infty$.*

We now make this concrete with some examples.

**Example 1 (Missing Mean Under Covariate Shift):** We begin with an example that will be familiar to machine learning practitioners. Let $Q$ be another distribution on $(X, Y)$. We assume that we observe samples of $X$ drawn from $Q$, but that $Y$ is unobserved. Our goal is to estimate the missing mean $\mathbb{E}_Q[Y]$. For example, if we collect health outcomes $(Y)$ in New York City $(P)$, our goal might be to use that data to infer average health outcomes in another city like Raleigh $(Q)$.

The issue is that New York and Raleigh are very different cities. But under the covariate shift assumption that $\mathbb{E}_P[Y|X] = \mathbb{E}_Q[Y|X]$, the missing mean can be written as:

$$\mathbb{E}_Q[Y] = \mathbb{E}_Q[\gamma_0(X)] = \mathbb{E}_P\left[\frac{dQ}{dP}(X)\gamma_0(X)\right] =: \theta_{\text{target}}(\gamma_0),$$

exactly as in Definition 1. In this example, Assumption 3 holds if and only if $Q$ is absolutely continuous with respect to $P$ and

$$\mathbb{E}_P\left[\frac{dQ}{dP}(X)^2\right] < \infty.$$

The analogous example in causal inference is estimating the counterfactual potential outcome for treated units when targeting the Average Treatment Effect on the Treated: $P$ are the control units, $Q$ are the treated units, and $Y(0)$ replaces $Y$. See Johansson et al. [2022] for discussion.

**Example 2 (Average Derivative):** We now consider an example common in applied economics. Let $X_1$ denote the first component of $X$. Then define the average derivative as:

$$\theta_{\text{target}}(\gamma_0) = \mathbb{E}_P\left[\frac{\partial\gamma_0(X)}{\partial X_1}\right].$$

This $\theta_{\text{target}}$ is also a linear functional. For example, if $Y$ is household spending, $X_1$ is household income, and the remainder of $X$ contains other household characteristics, then $\theta_{\text{target}}(\gamma_0)$ measures the average spending response to a change in income, known as the "Marginal Propensity to Consume."

A central object in what follows will be the *Riesz representer* corresponding to the estimand $\theta_{\text{target}}$:

**Definition 2** (Riesz representer). *Every continuous linear functional $\theta$ has a corresponding* Riesz representer, *a unique function $\alpha_\theta(x)$ such that:*

$$\theta(f) = \mathbb{E}_P[\alpha_\theta(X)f(X)],$$

*for all $f$ such that $\mathbb{E}_P[f(X)^2] < \infty$. We will write $\alpha_{target}(x)$ to denote the Riesz representer of $\theta_{target}$.*

**Example (Density Ratio):** When $\theta_{\text{target}}(f) = \mathbb{E}_Q[f(X)]$, then the Riesz representer is the density ratio, $\alpha_{\text{target}}(x) = dQ/dP(x)$. This has a known analytic form: $\alpha_{\text{target}}(x) = e(x)/(1 - e(x))$ where $e(x)$ is the propensity score or domain classifier for $Q$ vs. $P$.

Note that our setup focuses on scalar summaries of the optimal predictor, and does not, for example, consider finding a predictor that achieves small mean squared error uniformly over a target distribution $Q$. Recent work in Kern et al. [2024] suggests that we could extend our results to hold uniformly over $\mathcal{X}$. We leave such an extension to future work.

### 2.3 Plug-in estimation and regularization bias

Before turning to robustness and efficiency, we introduce a natural starting place, the *plug-in estimator*. This first fits $\hat{\gamma}(X)$ by predicting $Y$ from $X$ using samples from population $P$ and then computes:

$$\hat{\theta}_{\text{target}}(\hat{\gamma}) := \frac{1}{n_p}\sum_{i=1}^{n_p} m(\hat{\gamma}, X_i). \tag{1}$$

In the special case of the missing mean (Example 1), we fit our predictor under $P$, but apply it to covariates drawn from $Q$. That is, say that we observe $n_q$ iid samples of $X$ from $Q$. The plug-in estimate is then:[1] $\hat{\theta}_{\text{target}}(\hat{\gamma}) := \frac{1}{n_q}\sum_{j=1}^{n_q}\hat{\gamma}(X_j)$.

The core difficulty with the plug-in estimator is *regularization bias*. When fitting $\hat{\gamma}$ via machine learning, standard methods regularize the predictor to generalize better out-of-sample. Unfortunately, $\hat{\gamma}$ might regularize away parts of the sample-space that are important for $\theta_{\text{target}}$. Say there is a particular combination of $X$ that is very common in $Q$, but relatively rare in $P$. Then a cross-validated predictor trained under $P$ might regularize away the predictions on those values of $X$ to reduce variance. While optimal for prediction under $P$, this would lead to meaningful bias for $\mathbb{E}_Q[Y]$, which in turn could lead to a very poor estimate of the target estimand. Furthermore, bias means that the estimate will not be asymptotically normal, meaning that standard confidence intervals will not be valid — often an important desideratum in applied work.

### 2.4 Robustness: Constraining worst-case bias across many unknown targets

A large literature in robust optimization and algorithmic fairness focuses on constructing estimators that modify $\hat{\gamma}$ above such that the plug-in estimate $\hat{\theta}(\hat{\gamma})$ has small bias for a range of target quantities [Kim et al., 2022]. For example, say we observe data from a single source hospital $P$, but we want to estimate $\mathbb{E}_Q[Y]$ across many different target hospitals $Q_1, Q_2, \ldots Q_K$, where collecting unlabeled data and estimating density ratios $\alpha_\theta$ for each target site would be costly or possibly infeasible due to privacy concerns. Can we still estimate a $\hat{\gamma}$ from $P$ that is robust to many unknown distribution shifts? Kim et al. [2022] show that the answer is yes— although as we discuss below, we might be concerned about this procedure inflating the variance.

---

[1]We can technically write this as a special case of $m(\hat{\gamma}, X_i)$ over a single population by combining the two populations and introducing an indicator variable for membership in $Q$.

Consider a target estimand $\theta_{\text{target}}$ satisfying Definition 1 and Assumption 3, but now assume that we do not have access to $\theta_{\text{target}}$ ahead of time. Our goal is to use the observations in $P$ to construct a predictor $\hat{\gamma}$ such that the resulting plug-in estimator for $\theta_{\text{target}}$ is approximately unbiased. In other words, we want to control the worst-case bias over a large set of possible estimands. In the fair machine learning literature, this condition is known as *multiaccuracy*:

**Definition 3** (Multiaccuracy). *Given an "auditing" function class $\mathcal{C}$ and source population $P$, a predictor $f$ is $(\mathcal{C}, a)$-multiaccurate if for every function $c(X) \in \mathcal{C}$,*

$$\sup_{c \in \mathcal{C}} |\mathbb{E}_P [(Y - f(X)) \cdot c(X)]| \leq a.$$

Kim et al. [2019] show that an initial predictor $\hat{\gamma}_{\text{init}}$ can be modified to be multiaccurate by running a simple boosting procedure, including a version of our main proposal of boosting with ridge regression.

We now generalize the result in Kim et al. [2022], which considered the specific case of estimating a missing mean under covariate shift, to the more general class defined in Definition 1, where we replace the density ratio with the Riesz representer. Even if we do not know $\theta_{\text{target}}$ in advance, if we can construct a set $\Theta$ such that we believe $\theta_{\text{target}} \in \Theta$, then we can still obtain an unbiased estimator of $\theta_{\text{target}}$ if we can construct a multiaccurate predictor $\hat{\gamma}_{\text{ma}}$.

**Proposition 1.** *Let $\Theta$ be some set of functionals $\theta$ such that Definition 1 and Assumption 3 hold. Let $\mathcal{A}$ be the corresponding set of Riesz representers:*

$$\mathcal{A} := \{\alpha : \exists \theta \in \Theta \text{ s.t. } \theta(f) = \mathbb{E}[f(X)\alpha(X)], \forall f \text{ with } \mathbb{E}[f(X)^2] < \infty\}.$$

*Then if $\hat{\gamma}_{ma}$ is $(\mathcal{A}, a)$-multiaccurate:*

$$|\theta(\hat{\gamma}_{ma}) - \theta(\gamma_0)| \leq a, \forall \theta \in \Theta.$$

**Remark 1** (Distributionally-Robust Optimization). *Let $\Theta$ contain $\theta(f) = \mathbb{E}_{Q_k}[f(X)]$ for many distributions $Q_k$, where $\mathcal{A}$ contains the corresponding density ratios, $dQ_k/dP$. In this case, Definition 3 is a special case of the more general literature on Distributionally-Robust Optimization [Hastings et al., 2024]. For a target estimand, $\theta_{target}(f) = \mathbb{E}_{Q_{target}}[f(X)]$, if $dQ_{target}/dP \in \mathcal{A}$, then $\theta_{target}(\hat{\gamma}_{ma})$ is approximately unbiased for $\theta_{target}(\hat{\gamma}_0)$.*

However, we might be concerned about the cost of robustness in terms of additional variance. Since we enforce small bias over a potentially large class of target estimands $\theta \in \Theta$, we would therefore expect larger variance for our specific target estimand $\theta_{\text{target}}$.

## 2.5 Efficiency: Unbiased estimate with the smallest variance for a single, known target

In many applications, we know our target functional $\theta_{\text{target}}$ in advance, such as if we observe samples of $X$ from the distribution $Q$ at training time. In this case, one popular strategy is to "bias correct" the initial estimate $\hat{\theta}_{\text{target}}(\hat{\gamma})$, a problem studied extensively in the semiparametric statistics literature [Chernozhukov et al., 2024]. Importantly, the resulting estimator has the smallest possible variance among all unbiased estimators [Chernozhukov et al., 2018].

Following the general setup in Chernozhukov et al. [2021], we focus on bias correction using the Riesz representer of $\theta_{\text{target}}$. Under minimal conditions, if $\hat{\gamma}$ is a consistent estimator of $\mathbb{E}_P[Y|X]$, and $\hat{\alpha}$ is a consistent estimator of $\alpha_{\text{target}}(X)$, then the estimator,

$$\hat{\theta}_{\text{efficient}} := \hat{\theta}_{\text{target}}(\hat{\gamma}) + \underbrace{\frac{1}{n_p} \sum_{i=1}^{n_p} \hat{\alpha}(X_i)(Y_i - \hat{\gamma}(X_i))}_{\text{bias correction term}},$$

has the following three properties, asymptotically: (1) it is unbiased, i.e., $\mathbb{E}_P[\hat{\theta}_{\text{efficient}}] = \theta_{\text{target}}(\gamma_0)$; (2) it is normally distributed; and (3) it has the smallest variance of all regular asymptotically-linear estimators. This third property is called *semiparametric efficiency*, and the corresponding variance is called the *semiparametric efficiency bound*, denoted $V_\theta^*$ for $\theta(\gamma_0)$. Formally, we have:

$$\sqrt{n}\big(\hat{\theta}_{\text{efficient}} - \theta_{\text{target}}(\gamma_0)\big) \to \mathcal{N}(0, V_{\theta_{\text{target}}}^*), \text{ and } \hat{V} \to_p V_{\theta_{\text{target}}}^*,$$

where $\hat{V}$ is the sample variance,

$$\hat{V} := \frac{1}{n_p} \sum_{i=1}^{n_p} \left( m(\hat{\gamma}, X_i) + \hat{\alpha}(X_i)(Y_i - \hat{\gamma}(X_i)) - \hat{\theta}_{\text{efficient}} \right)^2.$$

See e.g. Chernozhukov et al. [2023] for a set of minimal conditions under which this result holds.

## 3 Ridge boosting simultaneously achieves robustness and efficiency

Thus far, we have explored two classes of estimators: robust estimators that have low bias over many estimands, and bias-corrected estimators that are unbiased and efficient for a specific target estimand. In this section, we demonstrate that it is possible to construct an estimator that is both robust and efficient. Specifically, when the set of target Riesz representers $\mathcal{A}$ is the norm ball in an RKHS, we can construct a multiaccurate predictor that has small worst-case bias over the corresponding $\Theta$ while simultaneously achieving the semiparametric efficient variance for *every* target $\theta \in \Theta$. We refer to the resulting procedure as *once-boosting with ridge regression* or, more simply, *ridge boosting*.

While our most general theoretical results only hold when $\mathcal{A}$ is a norm-ball in an RKHS, in the Appendix we sketch out a version of our result for boosting with Random Forests. Whether there exists a more general result is a exciting topic for future work.

### 3.1 Ridge boosting

In this section, we introduce the main estimator we analyze, once-boosting with ridge regression.

**Setup.** An RKHS $\mathcal{H}$ is a set of functions $h : \mathcal{X} \to \mathbb{R}$ defined by an inner product. In the most general case, for all $x \in \mathcal{X}$ there exists $\phi(x) \in \mathcal{H}$ such that for any $h \in \mathcal{H}$, $h(x) = \langle h, \phi(x) \rangle$. One special case is the finite dimensional Hilbert space where $\phi(x)$ is some feature map from $\mathcal{X} \to \mathbb{R}^d$ and $\mathcal{H} = \{h(x) = \beta^\top \phi(x) : \beta \in \mathbb{R}^d\}$; our results, however also hold for infinite-dimensional RKHS's. For any $h \in \mathcal{H}$, we define the norm $\|h\|_{\mathcal{H}}^2 = \langle h, h \rangle$.

For some RKHS $\mathcal{H}$, we consider the following function class:

$$\mathcal{A} = \{h \in \mathcal{H} : \|h\|_{\mathcal{H}} \leq B\},$$

for some $B > 0$. We set $B = 1$ (which will be without loss of generality), so that $\mathcal{A}$ forms a unit ball.

Recall that in our robustness setup, $\mathcal{A}$ corresponds to the set of Riesz representers for all $\theta \in \Theta$. In the covariate shift setting, $\mathcal{A}$ is the set of density ratios. Restricting our attention to Riesz representers that belong to such an $\mathcal{A}$ is very general, but not fully general. Such a set can include highly non-linear functions of $x$, but only functions that can be written in terms of the fixed basis $\phi(x)$. For example, $\mathcal{A}$ could be a set of functions that are linear in the last-layer embedding of a pre-trained large language model (LLM). But $\mathcal{A}$ could not include all functions achievable by fine-tuning that pre-trained LLM.

**Estimator.** We now introduce once-boosting with ridge regression. For notational simplicity, we present the algorithm in the case where $\mathcal{H}$ is a finite-dimensional RKHS with $\phi(x) \in \mathbb{R}^d$ — the arguments are identical in the infinite-dimensional case. We will write $\Phi_p \in \mathbb{R}^{n_p \times d}$ for the matrix with rows $\phi(x_i)$ for each observation $i$. Assume that we have an initial estimator of $\mathbb{E}_P[Y|X]$, $\hat{\gamma}_{\text{init}}(X)$, which could have been fit with some arbitrary machine learning algorithm. We then perform a ridge boosting step on the residuals $Y_p - \hat{\gamma}_{\text{init}}(X_p)$:

$$\min_{\beta \in \mathbb{R}^d} \left\{ \|Y_p - \hat{\gamma}_{\text{init}}(X_p) - \Phi_p\beta\|_2^2 + \lambda\|\beta\|_2^2 \right\}.$$

Call the solution $\hat{\beta}_{\text{boost}}$ and define $\hat{\gamma}_{\text{boost}}(x) := \phi(x)^\top \hat{\beta}_{\text{boost}}$. Then define:

$$\hat{\gamma}_{\text{ma}}(x) = \hat{\gamma}_{\text{init}}(x) + \hat{\gamma}_{\text{boost}}(x). \tag{2}$$

### 3.2 Ridge boosting is multiaccurate

Next we will show that $\hat{\gamma}_{\text{ma}}$ is indeed multiaccurate. In other words, $\theta(\hat{\gamma}_{\text{ma}})$ is an approxiamtely unbiased estimate of $\theta(\gamma_0)$, for all $\theta \in \Theta$. We first define the notion of the multiaccuracy error.

**Definition 4** (Multi-accuracy error). *For a given auditing function class $\mathcal{C}$ and a source population $P$, the multiaccuracy error of an estimator $\hat{f}(X)$ and its sample analog are defined as:*

$$MAE_{\mathcal{C}}(\hat{f}) = \sup_{c \in \mathcal{C}} |\mathbb{E}_P[c(X) \cdot (Y - \hat{f}(X))]|, \qquad \widehat{MAE}_{\mathcal{C}}(\hat{f}) = \sup_{c \in \mathcal{C}} |c(X_p)^\top (Y_p - \hat{f}(X_p))|.$$

Then we have the following guarantees:

**Theorem 1.** *For $\hat{\gamma}_{ma}$ defined in (2), we have:*

$$\widehat{MAE}_{\mathcal{A}}(\hat{\gamma}_{ma}) \leq \max_{1 \leq j \leq d} \frac{\lambda}{\lambda + \sigma_j^2} \widehat{MAE}_{\mathcal{A}}(\hat{\gamma}_{init}),$$

*where $\sigma_j^2$ are the eigenvalues of $\Phi_p^\top \Phi_p$. Under standard regularity conditions, with probability $1 - \eta$,*

$$MAE_{\mathcal{A}}(\hat{\gamma}_{ma}) \leq O\left(\delta_n + \sqrt{\frac{1/\eta}{n_p}}\right),$$

*for $\delta_n$ such that $\delta_n \to 0$ as $n \to \infty$.*

We provide a proof and additional discussion in the Appendix. The first result shows that one step of ridge boosting is guaranteed to decrease the sample multiaccuracy error. The second result shows we can generalize out of sample. The rate of convergence of $\delta_n$ depends on the dimensionality and smoothness of $\mathcal{H}$. When $\mathcal{H}$ is finite-dimensional with dimension $d$, $\delta_n \leq \sqrt{d/n}$.

**Remark 2.** *We emphasize that Theorem 1 is not a fundamentally new result. The multiaccuracy literature already proves generalization bounds on the multiaccuracy error for boosting estimators using ridge regression; see Kim et al. [2019]. However, our boosting procedure differs slightly in its specifics (e.g. linear boosting instead of exponential weighting) and so we provide Theorem 1 for completeness. Our proof uses standard techniques from the analysis of kernel ridge regression.*

### 3.3   Ridge boosting is semiparametrically efficient

We showed above that boosting with ridge regression produces a predictor that is multiaccurate with respect to $\mathcal{A}$. We now show that this multiaccurate estimator is also semiparametrically efficient for all $\theta \in \Theta$. Specifically, we will show that ridge boosting implicitly estimates the Riesz representer and performs semiparametric bias correction. In fact, the resulting estimator $\theta(\hat{\gamma}_{ma})$ is numerically equivalent to a special case of Automatic Debiased Machine Learning using kernel Riesz regression Chernozhukov et al. [2021], Singh [2024].

#### 3.3.1   Ridge regression implicitly estimates Riesz representers

The key fact that will lead to our main result is that ridge regression implicitly estimates Riesz representers. To see this, notice that for any continuous linear functional $\theta$, the Riesz representer $\alpha_\theta(X)$ is the unique solution to the following loss minimization problem

$$\alpha_\theta = \operatorname*{argmin}_{\alpha : \mathbb{E}[\alpha(X)^2] < \infty} \{\mathbb{E}_P[\alpha(X)^2] - 2\theta(\alpha)\}. \tag{3}$$

See Chernozhukov et al. [2021] for more discussion. One way to approximate $\alpha_\theta(X)$ is to minimize (3) over an RKHS $\mathcal{H}$. The sample version of this optimization problem is:

$$\min_{\eta \in \mathbb{R}^d} \left\{ \frac{1}{n_p} \eta^\top \Phi_p^\top \Phi_p \eta - 2\theta(\Phi_p)^\top \eta + \lambda \|\eta\|_2^2 \right\}, \tag{4}$$

with minimizer $\hat{\eta}_\lambda$ and corresponding Riesz representer estimate $\hat{\alpha}_\theta^\lambda(x) := \phi(x)^\top \hat{\eta}_\lambda$. See Singh [2024] for an analysis of this estimator.

Ridge regression when used to estimate $\theta$ can always be rewritten as a weighting estimator with weights $\hat{\alpha}_\theta^\lambda(X_p)$, as we show in the following proposition.

**Proposition 2.** *Let $\theta$ be any continuous linear functional, and define the Riesz representer estimate $\hat{\alpha}_\theta^\lambda$ from (4). Let $Z_p$ be any function of $X_p$ and $Y_p$. Consider the ridge regression in $\mathcal{H}$ that predicts $Z_p$ given $X_p$:*

$$\min_{\beta \in \mathbb{R}^d} \|Z_p - \Phi_p \beta\|_2^2 + \lambda \|\beta\|_2^2.$$

*Call the solution $\hat{\beta}_{ridge}$ and the corresponding predictor $\hat{\gamma}_{ridge}(x) = \phi(x)^\top \hat{\beta}_{ridge}$. Then:*

$$\hat{\theta}(\hat{\gamma}_{ridge}) = \frac{1}{n_p} \hat{\alpha}_\theta^\lambda(X_p)^\top Z_p.$$

This is a well-known result; see Kallus [2020], and see Bruns-Smith et al. [2025a] for an extensive discussion of the implications for semiparametric estimation.

**Remark 3.** *Ridge regression can be written as a linear smoother. The result above says that when we compute $\hat{\theta}(\hat{\gamma}_{ridge})$, the smoother weights estimate the Riesz representer. Random forests can also be written as a linear smoother, and Lin and Han [2022] show that the weights converge to the Riesz representer in a similar sense. We use this connection to show that the same robustness/efficiency properties apply to boosting with Random Forests in Appendix B.*

### 3.3.2 Main result: Semiparametric efficiency for all $\theta \in \Theta$

We now apply Section 3.3.1 to our estimator $\hat{\gamma}_{\mathrm{ma}}$ to establish our main result: $\theta(\hat{\gamma}_{\mathrm{ma}})$ does not just have small worst-case bias over all $\theta \in \Theta$, it is semiparametrically efficient for each individual $\theta \in \Theta$.

For any $\theta \in \Theta$, we have the following:

$$
\begin{aligned}
\hat{\theta}(\hat{\gamma}_{\mathrm{ma}}) &= \hat{\theta}(\hat{\gamma}_{\mathrm{init}}) + \hat{\theta}(\hat{\gamma}_{\mathrm{boost}}) \\
&= \hat{\theta}(\hat{\gamma}_{\mathrm{init}}) + \frac{1}{n_p}\hat{\alpha}_\theta^\lambda(X_p)^\top (Y_p - \hat{\gamma}_{\mathrm{init}}(X_p)),
\end{aligned} \tag{5}
$$

where the second equality follows from applying Proposition 2 for $Z_p = Y_p - \hat{\gamma}_{\mathrm{init}}(X_p)$.

Note that (6) has exactly the form of $\hat{\theta}_{\mathrm{efficient}}$ from Section 2.5. In fact, this estimation strategy — in which we fit an arbitrary machine learning estimator $\hat{\gamma}_{\mathrm{init}}$ and use a $\hat{\alpha}_\theta(X)$ that minimizes (3) for bias correction — is a well-studied estimator from the semiparametric statistics literature [Athey et al., 2018, Hirshberg and Wager, 2021, Chernozhukov et al., 2021, Bruns-Smith et al., 2025a]. The particular form of $\hat{\alpha}_\theta^\lambda(X)$ used here, which is obtained by minimizing (4) in an RKHS, is specifically considered in Hirshberg et al. [2019], Kallus [2020], Hazlett [2020], Singh [2024]. Thus, once-boosting with ridge provides a multiaccurate predictor, but the resulting point estimate $\theta(\hat{\gamma}_{\mathrm{ma}})$ is numerically-equivalent to well-studied semiparametrically efficient estimators. We leverage this connection to establish our main theoretical result.

**Theorem 2** (Informal). *Given standard regularity assumptions and some conditions on the quality of $\hat{\gamma}_{\mathrm{init}}$, then for all $\theta \in \Theta$, the ridge boosting plug-in estimator is asymptotically Normal and its variance achieves the asymptotic variance lower bound $V_\theta^*$:*

$$
\sqrt{n}\big(\hat{\theta}(\hat{\gamma}_{ma}) - \theta(\gamma_0)\big) \to \mathcal{N}(0, V_\theta^*), \text{ and } \hat{V} \to_p V_\theta^*.
$$

*where $\hat{V}$ is the sample variance,*

$$
\hat{V} := \frac{1}{n_p} \sum_{i=1}^{n_p} \big(m(\hat{\gamma}_{ma}, X_i) - \theta(\hat{\gamma}_{ma})\big)^2.
$$

See the Appendix for a formal Theorem statement and proof. This result establishes that for any estimand $\theta$ whose Riesz representer belongs to the RKHS ball $\mathcal{A}$, the ridge boosting estimator $\hat{\theta}(\hat{\gamma}_{\mathrm{ma}})$ is not just robust, it is semiparametrically efficient.

This estimator is also computationally convenient for practitioners interested in efficient inference. We simply take the initial predictor and run one step of boosting with kernel ridge regression, which has many readily-available implementations. And since a plug-in estimator using this new predictor is semiparametrically efficient for any estimand in $\Theta$, we do not have to (explicitly) fit individual Riesz representer estimates $\hat{\alpha}_\theta$ for each $\theta$, which can be expensive when there are many $\theta$s of interest. We also do not require specialized code for minimizing the Riesz loss (3), which can be a barrier for practitioners with less familiarity with Riesz representers.

## 4 Experiments

### 4.1 Simulation study

We now demonstrate in simulation that we simultaneously achieve robust and efficient inference by deploying our multiaccurate predictor on different estimands in different environments. To

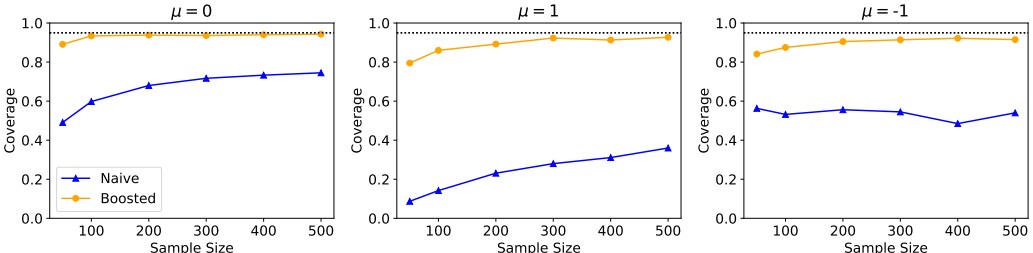

Figure 1: The empirical coverage comparisons between single kernel ridge outcome regression and kernel-ridge-boosted estimator across sample-sizes and the three environments $\mu \in \{0, 1, -1\}$. The blue lines with triangle markers plot the coverage for the base ridge model, and the orange line with circular markers for the once-boosted ridge model. The dotted line is at 0.95.

demonstrate the generality of the framework, we consider estimating an average derivative with correlated covariates — on its own, already a difficult task — under distribution shift. We will fit both ridge and once-boosted ridge models in a training sample, and then compute the usual 95% asymptotic Normal confidence interval for the average derivative across three test distributions, assessing empirical coverage over simulation draws.

**Simulation Setup:** We consider three-dimensional correlated covariates: $X_1, X_2 \sim N(\mu, 1)$, and

$$X_3 = 4 \cdot \sigma(X_1 - X_2) + \epsilon - 2,$$

where $\sigma(\cdot)$ is the sigmoid, and $\epsilon \sim N(0, 2^2)$. For the training distribution, $P$, $\mu = 0$; for the test distributions, we vary $\mu$. The outcomes are generated as:

$$Y = Y \sim f(X) + \eta, \quad f(X) = X_1 \cdot (0.2 + \sin(X_1) + \sigma(X_2) - 0.2 \cdot X_3), \quad \eta \sim N(0, 2^2).$$

The estimand is the average derivative of $f(X)$ with respect to $X_1$ under $Q$, $\mathbb{E}_Q[\partial f(X)/\partial X_1]$. We consider three different distributions for $Q$, with the same setup but with $\mu \in \{-1, 0, 1\}$. The dependence of $X_3$ on both $X_1$ and $X_2$ makes the average derivative more challenging to estimate.

**Methods:** We compare two estimators. (1) *Naive Kernel Ridge:* a standard kernel ridge outcome regression trained on the source data and plugged in for each target estimand. (2) *Boosted Kernel Ridge:* a one-step kernel ridge boosting procedure applied to the residuals of the initial kernel ridge regression. Both models are trained solely on the training (source) data and evaluated on each of the three test (target) distributions.

**Monte Carlo Simulation:** For each simulation, we draw a training sample of $X$ and $Y$ from the source distribution, and fit the kernel ridge and boosted kernel ridge model. We then draw one sample of $X$ from each of the three test distributions, and estimate the average derivative with respect to $X_1$ on that test distribution by symmetric differencing, along with the usual 95% asymptotic normal confidence interval. The whole process is repeated for sample sizes ranging from 50 to 500 and with 1,000 Monte Carlo replications. We report the empirical coverage of the confidence intervals for both methods across sample sizes and replications. The full simulation study is run on a four-core laptop.

**Results:** The results are shown in Figure 1. Across all three test distributions, the naive confidence intervals using the base ridge model under cover. By contrast, the confidence intervals from once-boosted ridge regression achieve good empirical coverage even with a moderate sample size ($n = 300$). The same pattern holds across all three randomly generated covariate shifts, showing the boosted ridge regression can achieve robustness toward covariate shifts and statistical efficiency at the same time. This reveals a new practical benefit of the multiaccurate estimator in this setting previously unexplored in the multicalibration literature: we achieve valid uncertainty quantification under distribution shift.

## 4.2 Empirical application to retirement income

We now consider an empirical economics application: estimating the age profile of income throughout retirement. For an individual $i$, let $Y_i$ be total income (including retirement income), let $A_i$ be age, and let $X_i$ be other covariates like education, race, and marital status. Define $\gamma_0(A, X) := \mathbb{E}[Y|A, X]$.

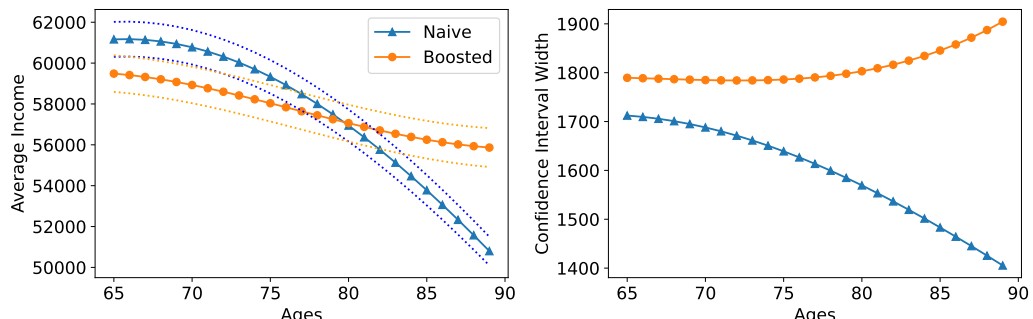

Figure 2: The left panel shows our point estimates for the age profile of income using a naive kernel ridge regression plug-in estimator and our ridge boosting estimator. The dotted lines are the corresponding 95% confidence intervals and the width of the intervals are plotted in the right panel.

Our estimand is the age profile of income (for ages $65, \dots, 89$), which is defined as the following *counterfactual* means:

$$\theta_{65} := \mathbb{E}[\gamma_0(65, X)], \quad \theta_{66} := \mathbb{E}[\gamma_0(66, X)], \quad \dots \quad , \quad \theta_{89} := \mathbb{E}[\gamma_0(89, X)].$$

That is, for each $\theta_a$, we want the average value of $\gamma_0(A, X)$, but where we have counterfactually replaced the age of every individual with the fixed value $a$. This is a key input into structural models of the macroeconomy — see, for example, Gourinchas and Parker [2002], Kaplan and Violante [2014]. Because the distribution of the covariates $X$ varies with $A$, this can induce very significant distribution shift, especially late in retirement. We emphasize that this is a highly simplified example inspired by Bruns-Smith et al. [2025b], although applying our methodology to their setting is a promising direction for future work.

State of the art modeling here would construct separate semiparametric efficient estimators for each point in the age profile, necessarily requiring a separate debiasing term at every age. In this application, we instead use a single multiaccurate predictor to estimate the 25 different target estimands—one for each year of retirement age— demonstrating the utility of our procedure in practice. We estimate the age profile of retirement using data from 2018 American Community Survey as processed by the `FolkTables` package [Ding et al., 2021]. As in Section 4.1 we fit a single kernel ridge and boosted kernel ridge model, and then compute plug-in estimates of $\theta_a$ using these two models. The results are displayed in Figure 2. Whereas the naive estimate (without boosting) features a steep decline of $11k from ages 65-89, our boosted estimates are substantially flatter — better matching the theoretical model for pension and social security income from Kaplan and Violante [2014]. Furthermore, while the naive confidence intervals shrink for the highest ages, our boosted confidence intervals actually grow slightly, suggesting that the naive model may undercover for the oldest part of the age profile.

## 5   Conclusion

In this manuscript, we investigate the connection between multiaccuracy and semiparametric efficiency. Specifically, we show that boosting an initial predictor with kernel ridge produces an estimator that is not only multiaccurate over estimands in an RKHS norm ball but is also semiparametrically efficient for each target separately. This result can be understood through the lens of Riesz regression: ridge boosting implicitly performs Riesz regression, thereby yielding an augmented balancing weight estimator that attains the semiparametric efficiency bound. We demonstrate one practical benefit: valid confidence intervals across distributions under covariate-shifts not seen at training time. However, our results are limited to shifts described by an RKHS. And making this proposal fully practical requires additional investigation of appropriate hyperparameter tuning. We hope this initial work leads to further exchange between the robustness and semiparametrics literatures.

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

# A  Proofs of Theoretical Results

## A.1  Proof of Proposition 1

**Proposition.** *Let $\Theta$ be some set of functionals $\theta$ such that Definition 1 and Assumption 3 hold. Let $\mathcal{A}$ be the corresponding set of Riesz representers:*

$$\mathcal{A} := \{\alpha : \exists \theta \in \Theta \text{ s.t. } \theta(f) = \mathbb{E}[f(X)\alpha(X)], \forall f \text{ with } \mathbb{E}[f(X)^2] < \infty\}.$$

*Then if $\hat{\gamma}_{ma}$ is $(\mathcal{A}, a)$-multiaccurate:*

$$|\theta(\hat{\gamma}_{ma}) - \theta(\gamma_0)| \leq a, \forall \theta \in \Theta.$$

*Proof.* For all $\theta \in \Theta$,

$$
\begin{aligned}
|\theta(\gamma_0) - \theta(\hat{\gamma}_{ma})| &= |\mathbb{E}_P[\alpha_\theta(X)\gamma_0(X)] - \mathbb{E}_P[\alpha_\theta(X)\hat{\gamma}_{ma}(X)]| \\
&= |\mathbb{E}[\alpha_\theta(X)(\gamma_0(X) - \hat{\gamma}_{ma}(X))]| \\
&= |\mathbb{E}[\alpha_\theta(X)(Y - \hat{\gamma}_{ma}(X))]| \\
&\leq \sup_{\alpha \in \mathcal{A}} |\mathbb{E}[\alpha(X)(Y - \hat{\gamma}_{ma}(X))]| \leq a.
\end{aligned}
$$

$\square$

## A.2  Proof of Theorem 1

**Assumption 2.** *We require the following boundedness conditions:*

1. *Assume that $\mathbb{E}_P[\|\phi(X)\|_{\mathcal{H}}^2] < B_1$.*

2. *Assume that $Y$, $\hat{\gamma}_{init}(X)$, and $\phi(X)$ are bounded $P$-almost-surely.*

**Theorem.** *For $\hat{\gamma}_{ma}$ defined in (2), we have:*

$$\widehat{MAE}_{\mathcal{A}}(\hat{\gamma}_{ma}) \leq \max_{1 \leq j \leq d} \frac{\lambda}{\lambda + \sigma_j^2} \widehat{MAE}_{\mathcal{A}}(\hat{\gamma}_{init}),$$

*where $\sigma_j^2$ are the eigenvalues of $\Phi_p^\top \Phi_p$. Under standard regularity conditions, with probability $1 - \eta$,*

$$MAE_{\mathcal{A}}(\hat{\gamma}_{ma}) \leq O\left(\delta_n + \sqrt{\frac{1/\eta}{n_p}}\right),$$

*for $\delta_n$ such that $\delta_n \to 0$ as $n \to \infty$.*

*Proof.* We begin by proving the first inequality — that the sample multiaccuracy of $\hat{\gamma}_{ma}$ is always smaller than that of $\hat{\gamma}_{init}$. We have:

$$\widehat{MAE}_{\mathcal{A}}(\hat{\gamma}_{ma}) = \sup_{a \in \mathcal{A}} \left| \frac{1}{n} a(X_p)^T(Y_p - \hat{\gamma}_{init}(X_p) - \Phi_p \hat{\beta}) \right|.$$

Here $\mathcal{A}$ is a Hilbert ball with norm 1. Since Hilbert spaces are self-dual, we can use the definition of the dual norm:

$$\widehat{MAE}_{\mathcal{A}}(\hat{\gamma}_{ma}) = \sup_{a \in \mathcal{A}} \left| \frac{1}{n} a(X_p)^T(Y_p - \hat{\gamma}_{init}(X_p) - \Phi_p \hat{\beta}) \right| = \frac{1}{n} \|\Phi_p^T(Y_p - \hat{\gamma}_{init}(X_p) - \Phi_p \hat{\beta})\|_2^2.$$

Similarly, the multi-accuracy error of $\hat{\gamma}_{init}$ is

$$\widehat{MAE}_{\mathcal{A}}(\hat{\gamma}_{init}) = \frac{1}{n} \|\Phi_p^T(Y_p - \hat{\gamma}_{init}(X_p))\|_2^2.$$

Using the closed-form for ridge regression, we have:

$$\hat{\beta}_{boost} = \left( \frac{1}{n} \Phi_p^T \Phi_p + \lambda I_d \right)^{-1} \frac{1}{n} \Phi_p^T(Y_p - \hat{\gamma}_{init}(X_p)).$$

Therefore:

$$\widehat{\mathrm{MAE}}_{\mathcal{C}}(\hat{\gamma}_{\mathrm{ma}}) = \frac{1}{n}\|\Phi_p^T(Y_p - \hat{\gamma}_{\mathrm{init}}(X_p)) - \Phi_p^T\Phi_p(\frac{1}{n}\Phi_p^T\Phi_p + \lambda I_d)^{-1}\frac{1}{n}\Phi_p^T(Y_p - \hat{\gamma}_{\mathrm{init}}(X_p))\|_2^2$$

$$= \frac{1}{n}\|(I_n - \frac{1}{n}\Phi_p^T\Phi_p(\frac{1}{n}\Phi_p^T\Phi_p + \lambda I_d)^{-1}) \cdot (Y_p - \hat{\gamma}_{\mathrm{init}}(X_p))\|_2^2.$$

Define

$$P_\lambda = \frac{1}{n}\Phi_p^T\Phi_p \left(\frac{1}{n}\Phi_p^T\Phi_p + \lambda I_d\right)^{-1}.$$

Consider the singular value decomposition (SVD) of $\frac{1}{\sqrt{n}}\Phi_p = U\Sigma_p^{1/2}V^T$, we have

$$P_\lambda = V\Sigma_p(\Sigma_p + \lambda I_d)^{-1}V^T$$

The eigenvalues of $P_\lambda$ are thus

$$\frac{\sigma_j^2}{\sigma_j^2 + \lambda}, \quad j = 1, \ldots, d,$$

where $\sigma_j^2$ are the eigenvalues of $\Phi_p^\top\Phi_p$. The eigenvalues of $I_n - P_\lambda$ are

$$\frac{\lambda}{\sigma_j^2 + \lambda}, \quad j = 1, \ldots, d.$$

Thus we have

$$\widehat{\mathrm{MAE}}_{\mathcal{A}}(\hat{\gamma}_{\mathrm{ma}}) \leq \frac{1}{n}\max_{1\leq j\leq d}\frac{\lambda}{\lambda + \sigma_j^2}\|\Phi_p^T(Y_p - \hat{\gamma}(X_p))\|_2^2 = \max_{1\leq j\leq d}\frac{\lambda}{\lambda + \sigma_j^2}\widehat{\mathrm{MAE}}_{\mathcal{A}}(\hat{\gamma}_{\mathrm{init}}).$$

Now we turn to the second inequality, that the multiaccuracy of $\hat{\gamma}_{\mathrm{ma}}$ generalizes. We apply the same dual norm argument to the population multiaccuracy error:

$$\mathrm{MAE}_{\mathcal{A}}(\hat{f}) = \sup_{a\in\mathcal{A}}|\mathbb{E}_P[a(X) \cdot (Y - \hat{\gamma}_{\mathrm{ma}}(X))]|$$

$$= \|\mathbb{E}_P[\phi(X) \cdot (Y - \hat{\gamma}_{\mathrm{ma}}(X))]\|_{\mathcal{H}}$$

$$\leq \sqrt{B_1\mathbb{E}_P[(Y - \hat{\gamma}_{\mathrm{ma}}(X))^2]}$$

$$= \sqrt{B_1\mathbb{E}_P[((Y - \hat{\gamma}_{\mathrm{init}}(X)) - \hat{\gamma}_{\mathrm{boost}}(X))^2]}$$

where the expression in the second line is well-defined when Assumption 4 holds, the third line uses Jensen's inequality for norms plus Cauchy-Schwarz and Assumption 4, and the last line uses the law of iterated expectations.

So ultimately, the expression depends on the mean squared error of kernel ridge regression for predicting $Y - \hat{\gamma}_{\mathrm{init}}(X)$ using $X$. Here we can use any off-the-shelf bound to get our main result.

For example, when $\phi(X) \in \mathbb{R}^d$, as we assume for simplicity in the main text, and if $\lambda$ is sufficiently small, we can apply the standard critical radius argument in Wainwright [2019] to get that with probability at least $1 - \eta$,

$$\mathbb{E}_P[((Y - \hat{\gamma}_{\mathrm{init}}(X)) - \hat{\gamma}_{\mathrm{boost}}(X))^2] \leq O\left(\sqrt{\frac{d}{n}} + \sqrt{\frac{1/\eta}{n}}\right).$$

More generally, if $\mathcal{H}$ is an infinite dimensional Hilbert space, then given an effective dimension $b$ and certain standard smoothness assumptions, then by following Fischer and Steinwart [2020], Singh [2024] we can show that taking $\lambda = (\log(n)/n)^{b/(b+1)}$, the squared generalization error of kernel ridge regression converges to 0 at rate:

$$O\left((\log(n)/n)^{b/(b+1)}\right).$$

$\square$

## A.3   Proof of Proposition 2 (Numerical equivalence of ridge regression and Riesz regression)

**Ridge Regression:** Given the auditing function class $\mathcal{H}$, ridge regression solves

$$\min_{\beta \in \mathbb{R}^d} \|Z_p - \Phi_p \beta\|_2^2 + \lambda \|\beta\|_2^2.$$

with closed-form solution

$$\hat{\beta}_\lambda = (\frac{1}{n_p} \Phi_p^T \Phi_p + \lambda I_d)^{-1} \cdot \frac{1}{n_p} \Phi_p^T Z_p.$$

The ridge boosting estimator is then given by

$$\theta(\Phi_p)^T \hat{\beta}_\lambda = \theta(\Phi_p)^T (\frac{1}{n_p} \Phi_p^T \Phi_p + \lambda I_d)^{-1} \cdot \frac{1}{n_p} \Phi_p^T Z_p.$$

**Riesz regression:** The Riesz regression solves

$$\hat{\eta}_\lambda = \min_{f \in \mathcal{F}} \left\{ \frac{1}{n_p} \eta^T \Phi_p^T \Phi_p \eta - 2\theta(\Phi_p)^\top \eta + \lambda \|\eta\|_2 \right\}.$$

The resulting Riesz representer estimator is

$$\hat{\alpha}_\theta^\lambda(\Phi_p) = \Phi_p \hat{\eta}_\lambda = \Phi_p (\frac{1}{n_p} \Phi_p^T \Phi_p + \lambda I)^{-1} \theta(\Phi_p).$$

The corresponding bias correction term is thus given by

$$\frac{1}{n_p} \hat{\alpha}_\theta^\lambda(\Phi_p)^T \cdot Z_p = \theta(\Phi_p)^T (\frac{1}{n_p} \Phi_p^T \Phi_p + \lambda I)^{-1} \cdot \frac{1}{n_p} \Phi_p^T Z_p.$$

This shows the numerical equivalence:

$$\frac{1}{n_p} \hat{\alpha}_\theta^\lambda(X_p)^T \cdot Z_p = \theta(\Phi_p)^T \hat{\beta}_\lambda.$$

Therefore, ridge regression estimator is numerically equivalent to the Riesz regression-based augmented estimator.

## A.4   Proof of Theorem 2

### A.4.1   Notation

We first introduce notation that will be used in the proofs.

- We denote the conditional excess risks of the nuisance estimators $\hat{\gamma}$ and $\hat{\alpha}$ given the observed covariates as follows:

$$R(\hat{\gamma}) := \mathbb{E}\left[ \{\hat{\gamma}(X) - \gamma_0(X)\}^2 \mid X_p \right], \quad R(\hat{\alpha}) := \mathbb{E}\left[ \{\hat{\alpha}(X) - \alpha_\theta(X)\}^2 \mid X_p \right].$$

- We denote the efficient influence function for $\theta(\gamma_0)$ as $\psi_0(X)$ and its high-order moments as

$$\sigma_0^2 = \mathbb{E}_Q[\psi_0(X)^2], \quad \kappa^3 = \mathbb{E}_Q[\psi_0(X)^3], \quad \zeta^4 = \mathbb{E}_Q[\psi_0(X)^4].$$

### A.4.2   Assumptions

**Assumption 3** (Continuity). *We assume that $\theta_{target}$ is a continuous linear functional. That is, there exists a constant $C > 0$ such that:*

$$\theta(f)^2 \le C\mathbb{E}_P[f(X)^2],$$

*for all $f$ with $\mathbb{E}[f(X)^2] < \infty$.*

**Assumption 4.** *We require the following boundedness conditions:*

1. *Assume that $\mathbb{E}_P[\|\phi(X)\|_{\mathcal{H}}^2] < B_1$.*

2. *Assume that $Y$, $\hat{\gamma}_{init}(X)$, and $\phi(X)$ are bounded $P$-almost-surely.*

**Assumption 5.** $\mathbb{E}_P\left[\gamma_0^2(X)\right] < \infty$ and $\sigma_0^2(X) = \mathbb{E}_P[(Y - \gamma_0(X))^2|X]$ is bounded.

**Assumption 6.** *The initial estimator $\hat{\gamma}$ for $\gamma_0$ is consistent in the sense that $R(\hat{\gamma}) \to_p 0$.*

**Assumption 7.** *The moments of the efficient influence function $\psi_0(X)$ satisfy*

$$\left(\left(\frac{\kappa}{\sigma}\right)^3 + \zeta^2\right) n^{-1/2} \to 0.$$

### A.4.3 Revisiting the theorems

**Theorem 3.** *Assume that we have a source population $P$ and Assumption 3-7 hold. Consider a RKHS $\mathcal{H}$ satisfying $\alpha_\theta \in \mathcal{H}$ and $\sqrt{n}R^{1/2}(\hat{\gamma}) \to_p 0$. .*

*Then the ridge boosting estimator with $\lambda = n^{-1/2}$ is asymptotically normal and its variance achieves the asymptotic variance lower bound $V_\theta^*$:*

$$\frac{\sqrt{n}(\hat{\theta}(\hat{\gamma}_{ma}) - \theta(\gamma_0))}{\hat{V}(\lambda)} \to \mathcal{N}(0,1), \text{ and } \hat{V}(\lambda) \to_p V_\theta^*.$$

*where*

$$\hat{V}(\lambda) = \frac{1}{n_Q}\sum_{i=1}^{n_Q}(\hat{\theta}(\hat{\gamma}_{init}) + \Phi_i^T(\Phi_p^T\Phi_p + \lambda I)^{-1}\Phi_p^T(Y_p - \hat{\gamma}_{init}(X_p)) - \hat{\theta}(\hat{\gamma}_{ma}))^2.$$

### A.4.4 Proof of Theorem 3

In Proposition 2, we established the numerical equivalence between ridge regression and Riesz regression. For any $\theta \in \Theta$, we have the following:

$$\begin{aligned}
\hat{\theta}(\hat{\gamma}_{ma}) &= \hat{\theta}(\hat{\gamma}_{init}) + \hat{\theta}(\hat{\gamma}_{boost}) \\
&= \hat{\theta}(\hat{\gamma}_{init}) + \bar{\Phi}_q^T\hat{\beta}_\lambda \\
&= \hat{\theta}(\hat{\gamma}_{init}) + \frac{1}{n_p}\hat{\alpha}_\theta^\lambda(X_p)^\top(Y_p - \hat{\gamma}_{init}(X_p)),
\end{aligned}$$

where the last equality follows from applying Proposition 2 for $Z_p = Y_p - \hat{\gamma}_{init}(X_p)$.

With this result, we then can derive the asymptotic normality using the Riesz regression formulation.

The asymptotic normality of the Riesz regression-based augmented estimator follows from Corollary 5.1 in [Chernozhukov et al., 2023] (Lemma 1 in the Appendix).

The constants $\bar{\sigma}$, $\bar{\alpha}$ and $\bar{\alpha}'$ are assumed to be bounded and are independent of sample size $n$. With a sufficiently large sample size, the last three conditions in Theorem 3 reduce to

$$\{R(\hat{\gamma})\}^{1/2} = o_p(1), \quad \{R(\hat{\alpha}_\lambda)\}^{1/2} = o_p(1), \quad \{nR(\hat{\gamma})R(\hat{\alpha}_\lambda)\}^{1/2} = o_p(1).$$

$\{R(\hat{\gamma})\}^{1/2} = o_p(1)$ is guaranteed from the conditions in Theorem 3.

Now it remains to study $\mathbb{E}\left[\{\hat{\alpha}_\lambda(X) - \alpha_\theta(X)\}^2 \mid X_p\right]$. Theorem 2 in [Singh, 2024] implies that when $\lambda = n^{-\frac{1}{2}}$,

$$\lim_{\tau \to \infty} \limsup_{n \to \infty} \mathbb{P}_{X_p \sim P}\left\{\mathbb{E}\left[\{\hat{\alpha}_\lambda(X) - \alpha_\theta(X)\}^2 \mid X_p\right] > \tau \cdot n^{-1}\right\} = 0.$$

Now we can show

$$\{R(\hat{\alpha}_\lambda)\}^{1/2} = o_p(1) \text{ and } \{nR(\hat{\gamma})R(\hat{\alpha}_\lambda)\}^{1/2} = o_p(1).$$

Therefore, we can directly show the asymptotic normality of the ridge boosting estimator.

$$\frac{\sqrt{n}(\hat{\theta}(\hat{\gamma}_{ma}) - \theta)}{\hat{V}(\lambda)} \to \mathcal{N}(0,1).$$

Finally, the consistency of the variance estimator follows from the law of large numbers.

### A.4.5 Additional Lemma

**Lemma 1** (Corollary 5.1 in [Chernozhukov et al., 2023]). *Suppose the following regularity and learning rate conditions hold as $n \to \infty$:*

- *The conditional variance is uniformly bounded:* $\mathbb{E}[(Y - \gamma_0(W))^2 \mid W] \le \bar{\sigma}^2$,

- *The nuisance parameter and estimator are uniformly bounded:* $\|\alpha_\theta\|_\infty \le \bar{\alpha}, \quad \|\hat{\alpha}\|_\infty \le \bar{\alpha}'$,

- *The moments of the efficient influence function are controlled:* $\left( \left( \frac{\kappa}{\sigma_0} \right)^3 + \zeta^2 \right) n^{-1/2} \to 0$,

- *The excess risks decay appropriately:*

$$
(\frac{\bar{\alpha}}{\sigma_0} + \bar{\alpha}') \cdot \{R(\hat{\gamma})\}^{1/2} = o_p(1), \quad \bar{\sigma} \cdot \{R(\hat{\alpha})\}^{1/2} = o_p(1), \quad \frac{\{nR(\hat{\gamma})R(\hat{\alpha})\}^{1/2}}{\sigma_0} = o_p(1).
$$

*Then the estimator $\hat{\theta}$ is consistent and asymptotically normal.*

$$
\hat{\theta} = \theta_0 + o_p(1), \quad \sigma_0^{-1} n^{1/2}(\hat{\theta} - \theta_0) \rightsquigarrow \mathcal{N}(0,1).
$$

## B   Random Forest Boosting

Our main "Universal Efficiency" argument for Ridge Boosting hinges on the fact that: For any $\theta \in \Theta$, we have the following:

$$
\begin{aligned}
\hat{\theta}(\hat{\gamma}_{\mathrm{ma}}) &= \hat{\theta}(\hat{\gamma}_{\mathrm{init}}) + \hat{\theta}(\hat{\gamma}_{\mathrm{boost}}) \\
&= \hat{\theta}(\hat{\gamma}_{\mathrm{init}}) + \frac{1}{n_p} \hat{\alpha}_\theta^\lambda(X_p)^\top (Y_p - \hat{\gamma}_{\mathrm{init}}(X_p)),
\end{aligned} \tag{6}
$$

which has a form of a debiased estimator. Following the semiparametric efficiency theory results from Chernozhukov et al. [2023], as long as $\hat{\gamma}_{\mathrm{init}}$ converges to $\gamma_0$ as $n \to \infty$, and $\hat{\alpha}_\theta^\lambda(X_p) \to \alpha_0$, and their product rate is sufficiently fast, the resulting estimator is efficient. Furthermore,

The implied weights that come from ridge boosting, $\hat{\alpha}_\theta^\lambda(X_p)$ are equivalent to a ridge minimizer of the Riesz loss. That allows us to use the fast rates proven in Singh [2024] for kernel Riesz representer estimates.

What about other linear smoothers that have the form: $\hat{\theta}(\hat{\gamma}_{\mathrm{linsmooth}}) = \frac{1}{n_p} w(X_p)^\top Z_p$? Ridge regression is one example with the important property that the smoother weights implicitly estimate the Riesz representer, but are there other examples?

Remarkably, Lin and Han [2022] show that Random Forests (which can be written as a linear smoother) also have implied weights that estimate the Riesz representer. Furthermore, the rate is fast enough to secure semiparametric efficiency. Therefore, boosting with Random Forests will have exactly the same efficiency guarantees that we show for ridge boosting. This is important because Random Forests are much more flexible than ridge regression in a fixed basis — indeed, one can think of Random Forests as kernel ridge regression but that learns an adaptive kernel. That would mean that the robustness/universal efficiency properties would hold to a wider class of estimands than just those expressible by a single RKHS.

However, the proof in Lin and Han [2022] is specific to ATE / covariate shift type estimands. It would be interesting to see if this could be generalized in future work to apply to generic Riesz representers.

## C   Connection between multiaccuracy and TMLE: A unified view from boosting

### C.1   Boosting to get a multiaccurate estimator

The multi-accurate estimator can be derived by a boosting strategy, where you first have an initial estimator $\hat{\gamma}$ of the outcome regression function, and then solve the following least square regression

problem:

$$\min_{\{\lambda_\phi\}_{\phi \in \mathcal{H}}} \mathbb{E}_P[(Y - \hat{\gamma}(X) - \sum_{\phi \in \mathcal{H}} \lambda_\phi \cdot \phi(X))^2].$$

Let $\{\hat{\lambda}_\phi\}_{\phi \in \mathcal{H}}$ be the optimal solution, the first order optimality condition for this is exactly the definition of multiaccuracy with $\alpha = 0$:

$$\mathbb{E}_P[h(X) \cdot (Y - \hat{\gamma}(X) - \sum_{\phi \in \mathcal{H}} \hat{\lambda}_\phi \cdot \phi(X))] = 0.$$

In this way, the final estimator $\hat{\gamma}(X) + \sum_{\phi \in \mathcal{H}} \hat{\lambda}_\phi \cdot \phi(X)$ is multiaccurate.

In the context of ridge boosting, you could treat the above $\mathcal{H}$ as the basis of the Hilbert space.

## C.2  TMLE

### C.2.1  TMLE can be viewed as boosting in the direction of the estimated clever covariate

The boosting step to get a multi-accurate estimator share a similar principle as the targeting step of the TMLE. In the targeting step of TMLE, we first have an initial estimator $\hat{\gamma}(X)$. Then, we specify a parametric working submodel:

$$\hat{\mu}_\epsilon(X) = \hat{\mu}(X) + \epsilon \cdot \hat{\phi}(X), \quad \epsilon \in (-\delta, \delta).$$

Here $\hat{\phi}(X)$ is the estimated clever covariate, which is usually given in the explicitly derived efficient influence function. Then the targeting step in TMLE solves the following boosting-type regression:

$$\hat{\epsilon} = \arg \min_\epsilon \mathbb{E}_P[(Y - \hat{\mu}_\epsilon(X))^2] = \arg \min_\epsilon \mathbb{E}_P[(Y - \hat{\mu}(X) - \epsilon \cdot \hat{\phi}(X))^2].$$

In the case of estimating ATE, solving the optimization once (one-step TMLE) will give you the semiparametrically efficient estimator under the standard assumptions, because it automatically solves the score equation

$$\mathbb{E}_P[\hat{\phi}(X) \cdot (Y - \hat{\mu}(X) - \hat{\epsilon} \cdot \hat{\phi}(X))] = 0.$$

and the update of clever covariate $\hat{\phi}$ is not needed.

## C.3  Ridge boosting multiaccurate estimator can be seen as a TMLE for multidimensional parameters with known clever covariate (Riesz representer)

### C.3.1  Ridge boosting multiaccurate estimator

Consider auditing with the function class,

$$\mathcal{C} = \left\{ c(x) = \beta^T \phi(x) : \|\beta\|_\mathcal{H} \leq B \right\} = \left\{ c(x) = \sum_{k=1}^d \beta_k \cdot \phi_k(x) : \|\beta\|_\mathcal{H} \leq B \right\},$$

We could implement the following boosting step:

$$\hat{\beta} := \arg \min_{\beta \in \mathbb{R}^d} \left\{ \mathbb{E}_P\left[\|y - \hat{\gamma}(X) - \phi(X)^T\beta\|_2^2\right] + \lambda \|\beta\|_2^2 \right\}$$

to obtain a multiaccurate estimator $\hat{\gamma}(X) + \phi(X)^T \hat{\beta}$.

### C.3.2  Viewing this as TMLE for multidimensional parameters

From the TMLE perspective, the auditing function class $\mathcal{C}$ here can be seen as a multidimensional (possibly infinite) parametric submodel in the targeting step. However, instead of targeting a single target parameter, we simultaneously target multiple parameters

$$[\mathbb{E}_{Q_1}(Y), \dots, \mathbb{E}_{Q_d}(Y)]^T.$$

with $dQ_k/dP = \phi_k(X)$, where $\phi_k(X)$ is the $k$th dimension basis of $\mathcal{C}$. Here $dQ_k/dP$ can be seen as the clever covariate for the target parameter $\mathbb{E}_{Q_k}(Y)$.

With this multidimensional parametric submodel, unregularized TMLE simulateneously solves multiple score equations for these different parameters.

$$\mathbb{E}_P[\phi_k(X) \cdot (y - \hat{\gamma}(X) - \phi(X)^T \hat{\beta})] = 0, \quad \forall k = 1, \dots, d.$$

By doing this, we get a simultaneously semiparametrically efficient estimator for

$$[E_{Q_1}(Y), \dots, E_{Q_d}(Y)]^T.$$

Moreover, if $dQ/dP$ ($Q$ is the target population) can be approximated by the linear combination of $[dQ_1/dP, \dots, dQ_d/dP]^T$. In other words,

$$\min_{c \in \mathcal{C}} \| \frac{dQ}{dP} - c \|_2 \to 0.$$

Then we could still obtain a semiparametrically efficient estimator for $\mathbb{E}_Q[Y]$ by applying $\hat{\gamma}(X) + \phi(X)^T \hat{\beta}$ on $X_q$ if the initial estimator $\hat{\gamma}$ is good enough in the sense that:

$$\|\hat{\gamma} - \gamma_0\|_2 \to 0, \text{ and } \sqrt{n}\|\hat{\gamma} - \gamma_0\|_2 \cdot \min_{c \in \mathcal{C}} \| \frac{dQ}{dP} - c \|_2 \to_p 0.$$

## D  Score-preserving TMLE

There is a recent paper about Score-Preserving TMLE [Pimentel et al., 2025]. The general idea is that instead of using the estimated clever covariate only, we add the basis functions in the nuisance parameter estimation into the targeting step. This paper mentioned that solving additional scores reduces the remainder term in the von-Mises expansion of our estimator because these scores may come close to spanning higher-order influence functions and result in an estimator with better finite-sample performance.

In the case of estimating $\mathbb{E}[Y(1)]$, the authors use the following parametric submodel in the targeting step:

$$\text{logit } \hat{Q}_n^{(\epsilon)}(W) = \text{logit } \hat{Q}_n(W) + \epsilon_0 \cdot \frac{A}{\hat{g}_n(W)} + \sum_j \epsilon_j \, \Phi_{n,j}(W),$$

where $\hat{Q}_n(W)$ is the initial estimator, $\frac{A}{\hat{g}_n(W)}$ is the estimated clever covariate and $\Phi_{n,j}(W)$ is the basis function in the nuisance parameter estimation. There is a deep connection between Score-preserving TMLE and ridge boosting: If we use ridge outcome regression and ridge boosting with function class $\mathcal{C}$, it seems that we are indeed performing a version of score-preserving TMLE because we include the basis functions of outcome regression in this submodel, and the clever covariate can also be represented using these basis functions. Right now, the Score-preserving TMLE is studied in the setting of highly adaptive lasso. But there is a recent paper on highly adaptive ridge [Schuler et al., 2024], where ridge regression might be a good fit for score-preserving TMLE.

