# OpenReview forum: "Ridge Boosting is Both Robust and Efficient"
_NeurIPS.cc/2025/Conference — NeurIPS 2025 spotlight_

### Official Review · Reviewer_wM69 · 2025-07-02

**Clarity:** 4
**Significance:** 3
**Originality:** 3
**Rating:** 5
**Confidence:** 5

**Summary:**

The paper considers the problem of estimating the mean outcome $\mathbb{E}_Q[Y],$ in a target population $Q$, denoted given labeled data $(X_i, Y_i)$ from a source distribution $P$. The authors adopt the \emph{target-independent learning} setup introduced by Kim et al. (2022). In particular, Kim et al. (2022) establish an interesting connection between multi-accurate predictors (a concept in fair machine learning) and this distribution shift problem, showing that multi-accurate predictors are ``universally adaptable'' meaning that they achieve low bias for any target distribution $Q$ for which the auditing class $\mathcal{C}$ that well approximates $dQ(x) / dP(x)$.

This paper expands upon the connections between multi-accurate predictors and causal inference. The main result establishes a numerical equivalence: when the auditing class $\mathcal{C}$ is a Hilbert ball, then the multiaccurate predictor can be implemented as a boosted ridge regression $\hat{\psi}(X) = \hat{\gamma}(X) + X^\top \hat{\beta}$ and consequently it is numerically equivalent to the automatic debiased machine learning estimator (e.g., Chernozhukov et al. 2022). Consequently, the multi-accurate predictor is not only universally adaptable, but it is also asymptotically normal for any target distribution and semi-parametrically efficient. One practical upshot of this connection is that we can construct valid confidence intervals for $\mathbb{E}_Q[Y]$ based on this multi-accurate predictor using standard techniques as illustrated in their Monte Carlo simulations.

**Questions:**

Please see my earlier discussion of strengths/weaknesses of the paper.

**Ethical Concerns:**

["NO or VERY MINOR ethics concerns only"]

**Final Justification:**

The paper provides a nice connection between multi-accuracy and semi-parametric statistics. The authors addressed my main concerns during the rebuttal.

**Limitations:**

Please see my earlier discussion of strengths/weaknesses of the paper.

**Paper Formatting Concerns:**

I have no concerns about the paper formatting.

**Quality:**

4

**Strengths And Weaknesses:**

I found the paper to be very well-written, effectively presenting its main result and describing how it builds upon the connections between fair machine learning and causal inference that were initiated in Kim et al. (2022). The paper's main result is ex-post obvious once it is shown, but I certainly had not noticed this nice connection before despite having familiarity with both multi-accuracy and the auto-DML framework. In this sense, the paper provides a nice bridge between fair machine learning and causal inference/semi-parametric theory, which have two active areas at the intersection of statistics and machine learning.

I have one main concern / point of clarification for the authors' main result. The central assumption in the paper is that the density ratio $dQ/dP$ can be well approximated by a linear function of the covariates $X$ (the first condition in Theorem 2). However, as stated in the main text, this assumption appears slightly stronger than those typically made in the Auto-DML literature (e.g., Chernozhukov et al. 2022). In particular, the Auto-DML framework typically assumes that the density ratio lies in the span (or is well approximated) by a (possibly growing) class of nonlinear transformations of the covariates $\frac{dQ}{dP}(x) \approx \sum_{j=1}^K \delta_j \phi_j(x), \quad \phi_j \in \mathcal{F}$, where $\mathcal{F}$ is a dictionary of basis functions (e.g., polynomials, splines, neural networks). I was thinking through the simple example where $X \sim N(0, 1)$ on $P$ and $X \sim N(1, 1)$ on $Q$ in which case the density ratio is $dQ / dP(x) = \exp(x - 0.5)$, which is clearly not linear in $x$.

Of course, the feature space $X$ in this paper could be re-interpreted in a broader view -- for example, we could think of $X$ as already the non-linear bases typically used in the Auto-DML framework. Indeed, I looked in the Appendix and the authors already noted this, and so I would strongly encourage them to move this into the main text as a remark. Of course, this resolution also highlights a limitation of this connection --- for this construction, the researcher must pre-commit to the feature map ex-ante, rather than trying to learn a sufficiently expressive representation.

---

> ### Author Rebuttal · Authors · 2025-07-30
>
> # General Response
>
> ## Expanded Results:
>
> We appreciate reviewers' thoughtful engagement with our submission. To streamline exposition in our initial submission, we characterized the results in terms of linear functions of $X$ and focused on the estimand $\mathbb{E}_Q[Y]$. In revisiting our technical results, we realized that we can substantially generalize both the class of functions and target estimands we consider --- with essentially no changes to the proofs at all. We believe that these revised results are responsive to reviewer questions and greatly expand the scope of our paper.
>
> * **Beyond linearity in $X$.** First, our results hold exchanging linear functions of $X$ for linear functions in features $\phi(X)$, which is a remarkably broad class of functions. This class includes when $\phi$ are the infinite-dimensional features from an RKHS; in this case the resulting algorithm would be once-boosted kernel ridge regression. This class also includes adaptive bases that are selected via sample splitting, such as honest random forests and ridge regression on the last-layer embedding of a pre-trained neural network.
>
>      An important limitation is that the results using our proof techniques are not fully general and do not hold for adaptive bases without sample splitting. That said, there are other examples where boosting also implicitly estimates a density ratio that do not rely on ridge regression or sample splitting. One prominent instance is random forests, which Lin and Han [1] show implicitly estimate a density ratio. In our revised manuscript, we adapt their results to show that, under certain restrictions, random forests satisfy the requirements in our Theorem 2; and therefore that boosting with random forests enjoys the same properties as boosting with ridge. While we leave a thorough discussion of this to future work, we think this is a very exciting direction that opens up many promising research avenues.
>
>
> * **More general estimands** Second, we show that our results continue to hold when we replace the estimand $\mathbb{E}_Q[Y]$ with any linear functional of $\mathbb{E}[Y|X]$, including average derivatives, average impulse responses, average treatment effects and more. To do so, we modify the class $\mathcal{C}$ to contain the Riesz representers for these functionals: a special case is the density ratio $dQ/dP$, which is the Riesz representer for $\mathbb{E}_Q[Y]$. Again, we achieve this generality without changing the proofs.
>
>      One important upshot of this generalization is that we can simultaneously consider a vector of estimands if each has a Riesz representer in the set $\mathcal{C}$. Many applied problem estimate a vector of estimands, and existing semiparametric estimators require fitting a separate debiasing model for each individual point in the vector. Our once-boosted ridge procedure only requires fitting a single model, which can then be used to construct the whole vector of estimates. This is practically very appealing, although a thorough assessment of this is beyond the scope of the current paper, especially with respect to valid joint inference. In particular, in follow up work we will better articulate conditions under which the 95\% confidence band over the whole vector is simultaneously valid.
>
> [1] Lin, Zhexiao, and Fang Han. "On regression-adjusted imputation estimators of the average treatment effect." arXiv preprint arXiv:2212.05424 (2023).
>
> ## Difficult Average Derivative Simulation:
>
> To demonstrate our results in a more general setting, we consider a non-linear simulation where we estimate the average derivative across multiple distributions (combining distribution shift and a new estimand). For these initial results, we retain a synthetic setup because we would like to show coverage of the confidence intervals. We choose a setting with 3 covariates: $X_1, X_2$ are iid $N(\mu,1)$, $X_3 = 4 \cdot \sigma(X_1 - X_2) + \epsilon - 2$, where $\sigma(\cdot)$ is the sigmoid, and $\epsilon \sim N(0,2)$. For the training distribution $\mu = 0$.
>
> The outcomes follow: $Y \sim f(X) + \eta$, where $f(X) = X_1 \cdot( 0.2 + \sin(X_1) + \sigma(X_2) - 0.2 \cdot X_3)$ and $\eta \sim N(0,2)$. The estimand is the average derivative under $Q$, $\mathbb{E}_Q[ \partial f(X) / \partial X_1 ]$. We consider three different distributions for $Q$. Each is identical to the setup above but changes the value of $\mu \in \{-1,0,1\}$. The dependence of $X_3$ on both $X_1$ and $X_2$ makes the average derivative very difficult to estimate, as we will show.
>
> We fit kernel ridge regression (with an RBF kernel) of $Y$ on $X$ in the training set. Then we also fit a once-boosted kernel ridge regression on the training set. For each we compute estimates of the average derivative of the model under the three Q distributions. The nominal (not boosted) krr estimate *badly* undercovers: with $n = 500$, the $95\\%$ coverage for the three target estimands is $49\\%$, $70\\%$, and $31\\%$ respectively. Our boosted estimator achieves coverage $93\\%$, $95\\%$, $92\\%$ respectively. In this challenging non-linear setting the gap in performance is actually much larger than in the linear ridge setting we originally submitted. (The final submission will include full figures for this experiment, which we cannot include in the rebuttal.)
>
> ## Real Data Experiment
>
> Following reviewer feedback, we also applied this approach to a real data example using American Community Survey (ACS) data, where the goal is to estimate counterfactual outcome means across a vector of ages. In this setting, $Y$ is income, $A$ is age, and $X$ are covariates like occupation, marriage status, and place of birth. Let $f(A,X) = \mathbb{E}[Y|A,X]$. Our estimands are $\mathbb{E}[f(65, X)], \mathbb{E}[f(66, X)], ... \mathbb{E}[f(90, X)]$. This vector of estimands represents the "age profile" of total income in retirement keeping $X$ fixed. This profile is an important input in many macroeconomic models (see, for example, the seminal paper from Kaplan and Violante [2]), especially for modeling consumption in retirement.
> Each of these estimands has substantial covariate shift when focusing on ages over 65.
> Typical modeling here would construct separate semiparametric efficient estimators for each point in the age profile, necessarily requiring a separate debiasing term at every age.
> We can instead use our theoretical results to motivate a common debiasing procedure for the entire vector of estimands: in our example we fit kernel ridge regression to estimate $f(A,X)$ and then use one round of boosting. Finally, we use the boosted $\hat{f}$ to compute a semiparametric-efficient estimate of every point on the age profile with corresponding confidence intervals.
>
> | Age | Naive Estimate | Boosted Estimate | Naive C.I. Width | Boosted C.I. Width |
> | --- | -------------- | ---------------- | ---------------- | ------------------ |
> | 65 | 61.1k | 59.5k | 1.7k | 1.8k |
> | 71 | 60.6k | 58.8k | 1.7k | 1.8k |
> | 77 | 58.5k | 57.6k | 1.6k | 1.8k |
> | 83 | 55.1k | 56.5k | 1.5k | 1.8k |
> | 89 | 50.8k | 55.9k | 1.4k | 1.9k |
>
> We give the results (in thousands of US dollars) for 5 evenly-spaced points on the age profile in the table above. Whereas the naive estimate (without boosting) features a steep decline of \$11k from ages 65-89, our boosted estimates are substantially flatter --- better matching the theoretical model for pension and social security income from Kaplan and Violante. Furthermore, while the naive confidence intervals shrink for the highest ages, our boosted confidence intervals actually grow slightly, suggesting that the naive model may undercover for the oldest part of the age profile.
>
> [2] Kaplan, Greg, and Giovanni L. Violante. "A model of the consumption response to fiscal stimulus payments." Econometrica 82.4 (2014): 1199-1239.
>
> # Reviewer-Specific Response
>
> Thank you for your very helpful feedback on our submission. Following your advice, we have updated the main text to work in a basis expansion $\phi(X)$, which could possibly be infinite dimensional. We think this has made the paper much stronger, and many other reviewers were also concerned about modeling density ratios as linear. In fact, we originally used the multivariate Gaussian example with different means in the main text --- so like you say, this is technically a mis-specified example (and maybe this explains why we actually over-cover slightly in Figure 1). Our new simulation setting with kernel ridge boosting avoids this mis-specification angle. Thank you for pointing this out!
>
> When applying our ridge-regression-specific arguments, we believe that your observation highlights the core split between when our results hold or not: the basis must be chosen in advance. This can include representations that are learned in separate samples, but then we must run once-boosted ridge regression on that learned representation.
>
> Interestingly, while revising our main argument to apply to basis expansions $\phi(X)$, we noticed that the Lin and Han (2023) paper mentioned in the general response shows a new avenue for achieving the same result we show in Theorem 2 but via once-boosting with a random forest. In practice, researchers in the multicalibration literature seem to prefer auditing with tree classes, and so this is a possibly important generalization for connecting the two literatures. Their argument involves characterizing random forests as a linear smoother, and shows that the implied weights of that smoother implicitly approximate the density ratio from the training distribution to the test distribution on which the random forest is applied. In future work we hope to explore whether or not the argument holds for more general classes of linear smoothers.
>
> Again, we are grateful for your thoughtful engagement with our paper and believe that the revised version is stronger as a result.

---

### Official Review · Reviewer_6txf · 2025-07-03

**Clarity:** 1
**Significance:** 3
**Originality:** 2
**Rating:** 4
**Confidence:** 2

**Summary:**

This paper studies the problem of estimating $\mathbb{E}_Q[Y]$ from labeled samples $(X_i, Y_i) \sim P$ under covariate shift, where $\mathbb{E}_P[Y|X] = \mathbb{E}_Q[Y|X]$. The goal is to construct an estimator that generalizes well to the target distribution.

The main idea of this work is to apply multiaccuracy: a predictor $\hat{f}$ is said to be $(\mathcal{C}, \alpha)$-multiaccurate if
$$
\sup_{c \in \mathcal{C}} \left| \mathbb{E}_P\left[ (Y - \hat{f}(X)) \cdot c(X) \right] \right| \leq \alpha.
$$

If $dQ/dP \in \mathcal{C}$, then this guarantees that the expected bias  $\mathbb{E}_Q[Y - \hat{f}(X)]$ is at most $\alpha$.

The estimator studied in the paper is a ridge-boosted predictor of the form
$$
\hat{\psi}(X) = \hat{\gamma}(X) + X^\top \hat{\beta},
$$
where $\hat{\gamma}$ is any initial predictor trained on $(X_p, Y_p)$ and $\hat{\beta}$ is obtained by ridge regression on the residuals $Y - \hat{\gamma}(X)$ using source data.

**Theorem 1** shows that this ridge-boosted predictor has lower multiaccuracy error than the initial predictor. That is, the worst-case residual correlation with functions in $\mathcal{C}$ is reduced by the boosting step. In the case where the $P$ has a diagonal covariance matrix, they can show that multiaccuracy tends to 0 as the regularization, $\lambda$ goes to 0.

In their main result, **Theorem 2**, they  show that the average prediction of the ridge-boosted estimator over target samples is asymptotically normal and achieves the semiparametric efficiency bound for estimating $\mathbb{E}_Q[Y]$, provided that $\hat{\gamma}$ is consistent and $dQ/dP$ is well-approximated in $\mathcal{C}$.

Finally, they experimentally validate their results over synthetic data.

**Questions:**

Please see the list of weaknesses I provided. In general, I think this paper could do with significant revision. It also needs to justify its setting (where $\mathcal{C}$ is just a Hilbert ball more).

**Ethical Concerns:**

["NO or VERY MINOR ethics concerns only"]

**Final Justification:**

See comment to authors.

**Limitations:**

Yes.

**Paper Formatting Concerns:**

None.

**Quality:**

2

**Strengths And Weaknesses:**

Strengths: this work studies an interesting connection between multiaccuracy and OOD generalization. It also provides a relatively thorough analysis of the proposed algorithm (showing that the estimator achieves the lower bound on variance) which is a nice guarantee.

Weaknesses:

1. The setting where $\mathcal{C}$ only comprises of linear functions is rather limited. It doesn't feel very plausible to me that dQ/dP would follow this form in almost any situation. If the authors don't agree here, I would find some examples very helpful. At the very least, arguing why this is an important theoretical step would be appreciated.

2. Theorem 1 feels a bit weak. Merely showing that the multiaccuracy is 'smaller' than the previous one doesn't feel like much of a result on its own. This is only helpful in the context of multiaccuracy being a goal (from some kind of fairness perspective). However, for the OOD setting (where the goal is to just achieve a good loss over Q) I don't really see how this is relevant.

3. Theorem 2 feels like a relatively straightforward consequence of the Corollary from Chernozhukov et. al. 2023. This wouldn't be an issue if Theorem 2 were characterizing a more relevant or practical problem, but given that Theorem 2 is restricted to a very limited theoretical setting (weakness 1) i do feel like there is a higher onus for providing new technical innovatinos.

4. The presentation of Theorem 2 is sloppy. what does $\min_{c \in C} ||\frac{dQ}{dP} - c|| \to_p 0$ mean? What are we taking the limit with respect to? None of those quantities vary with the sample size. Furthermore, insepcting the proofs, it seems that this work effectively just plugs this in all over the place to achieve other limits going to 0. What this means is that simply stating $\frac{dQ}{dP} \in \mathcal{C}$ would have equivalently sufficed. As far as I can tell, $\frac{dQ}{dP} \in \mathcal{C}$ is the only reasonable interpretation of this statement, but I'm happy to be corrected.

5. The general presentation can be significantly improved. There are several passages where algebra is done to provide intuition, but I think that this can be mostly omitted -- the manipulations are relatively standard and mostly take up space. Instead, this space could be used to argue more effectively for the relevance of this setting and to provide more proof details. The supplementary material of this paper is quite short suggesting that some of the proofs could be included in the main body.

6. Additionallly the notation feels cumbersome and often poorly defined. For example, $\Psi(Q)$ is never defined and appears for the first time in Theorem 2. I'm assuming that $\Psi(Q)$ simply means $\mathbb{E}_Q[Y|X]$.

---

> ### Author Rebuttal · Authors · 2025-07-30
>
> # General Response
>
> ## Expanded Results:
>
> We appreciate reviewers' thoughtful engagement with our submission. To streamline exposition in our initial submission, we characterized the results in terms of linear functions of $X$ and focused on the estimand $\mathbb{E}_Q[Y]$. In revisiting our technical results, we realized that we can substantially generalize both the class of functions and target estimands we consider --- with essentially no changes to the proofs at all. We believe that these revised results are responsive to reviewer questions and greatly expand the scope of our paper.
>
> * **Beyond linearity in $X$.** First, our results hold exchanging linear functions of $X$ for linear functions in features $\phi(X)$, which is a remarkably broad class of functions. This class includes when $\phi$ are the infinite-dimensional features from an RKHS; in this case the resulting algorithm would be once-boosted kernel ridge regression. This class also includes adaptive bases that are selected via sample splitting, such as honest random forests and ridge regression on the last-layer embedding of a pre-trained neural network.
>
>      An important limitation is that the results using our proof techniques are not fully general and do not hold for adaptive bases without sample splitting. That said, there are other examples where boosting also implicitly estimates a density ratio that do not rely on ridge regression or sample splitting. One prominent instance is random forests, which Lin and Han [1] show implicitly estimate a density ratio. In our revised manuscript, we adapt their results to show that, under certain restrictions, random forests satisfy the requirements in our Theorem 2; and therefore that boosting with random forests enjoys the same properties as boosting with ridge. While we leave a thorough discussion of this to future work, we think this is a very exciting direction that opens up many promising research avenues.
>
>
> * **More general estimands** Second, we show that our results continue to hold when we replace the estimand $\mathbb{E}_Q[Y]$ with any linear functional of $\mathbb{E}[Y|X]$, including average derivatives, average impulse responses, average treatment effects and more. To do so, we modify the class $\mathcal{C}$ to contain the Riesz representers for these functionals: a special case is the density ratio $dQ/dP$, which is the Riesz representer for $\mathbb{E}_Q[Y]$. Again, we achieve this generality without changing the proofs.
>
>      One important upshot of this generalization is that we can simultaneously consider a vector of estimands if each has a Riesz representer in the set $\mathcal{C}$. Many applied problem estimate a vector of estimands, and existing semiparametric estimators require fitting a separate debiasing model for each individual point in the vector. Our once-boosted ridge procedure only requires fitting a single model, which can then be used to construct the whole vector of estimates. This is practically very appealing, although a thorough assessment of this is beyond the scope of the current paper, especially with respect to valid joint inference. In particular, in follow up work we will better articulate conditions under which the 95\% confidence band over the whole vector is simultaneously valid.
>
> [1] Lin, Zhexiao, and Fang Han. "On regression-adjusted imputation estimators of the average treatment effect." arXiv preprint arXiv:2212.05424 (2023).
>
> ## Difficult Average Derivative Simulation:
>
> To demonstrate our results in a more general setting, we consider a non-linear simulation where we estimate the average derivative across multiple distributions (combining distribution shift and a new estimand). For these initial results, we retain a synthetic setup because we would like to show coverage of the confidence intervals. We choose a setting with 3 covariates: $X_1, X_2$ are iid $N(\mu,1)$, $X_3 = 4 \cdot \sigma(X_1 - X_2) + \epsilon - 2$, where $\sigma(\cdot)$ is the sigmoid, and $\epsilon \sim N(0,2)$. For the training distribution $\mu = 0$.
>
> The outcomes follow: $Y \sim f(X) + \eta$, where $f(X) = X_1 \cdot( 0.2 + \sin(X_1) + \sigma(X_2) - 0.2 \cdot X_3)$ and $\eta \sim N(0,2)$. The estimand is the average derivative under $Q$, $\mathbb{E}_Q[ \partial f(X) / \partial X_1 ]$. We consider three different distributions for $Q$. Each is identical to the setup above but changes the value of $\mu \in \{-1,0,1\}$. The dependence of $X_3$ on both $X_1$ and $X_2$ makes the average derivative very difficult to estimate, as we will show.
>
> We fit kernel ridge regression (with an RBF kernel) of $Y$ on $X$ in the training set. Then we also fit a once-boosted kernel ridge regression on the training set. For each we compute estimates of the average derivative of the model under the three Q distributions. The nominal (not boosted) krr estimate *badly* undercovers: with $n = 500$, the $95\\%$ coverage for the three target estimands is $49\\%$, $70\\%$, and $31\\%$ respectively. Our boosted estimator achieves coverage $93\\%$, $95\\%$, $92\\%$ respectively. In this challenging non-linear setting the gap in performance is actually much larger than in the linear ridge setting we originally submitted. (The final submission will include full figures for this experiment, which we cannot include in the rebuttal.)
>
> ## Real Data Experiment
>
> Following reviewer feedback, we also applied this approach to a real data example using American Community Survey (ACS) data, where the goal is to estimate counterfactual outcome means across a vector of ages. In this setting, $Y$ is income, $A$ is age, and $X$ are covariates like occupation, marriage status, and place of birth. Let $f(A,X) = \mathbb{E}[Y|A,X]$. Our estimands are $\mathbb{E}[f(65, X)], \mathbb{E}[f(66, X)], ... \mathbb{E}[f(90, X)]$. This vector of estimands represents the "age profile" of total income in retirement keeping $X$ fixed. This profile is an important input in many macroeconomic models (see, for example, the seminal paper from Kaplan and Violante [2]), especially for modeling consumption in retirement.
> Each of these estimands has substantial covariate shift when focusing on ages over 65.
> Typical modeling here would construct separate semiparametric efficient estimators for each point in the age profile, necessarily requiring a separate debiasing term at every age.
> We can instead use our theoretical results to motivate a common debiasing procedure for the entire vector of estimands: in our example we fit kernel ridge regression to estimate $f(A,X)$ and then use one round of boosting. Finally, we use the boosted $\hat{f}$ to compute a semiparametric-efficient estimate of every point on the age profile with corresponding confidence intervals.
>
> | Age | Naive Estimate | Boosted Estimate | Naive C.I. Width | Boosted C.I. Width |
> | --- | -------------- | ---------------- | ---------------- | ------------------ |
> | 65 | 61.1k | 59.5k | 1.7k | 1.8k |
> | 71 | 60.6k | 58.8k | 1.7k | 1.8k |
> | 77 | 58.5k | 57.6k | 1.6k | 1.8k |
> | 83 | 55.1k | 56.5k | 1.5k | 1.8k |
> | 89 | 50.8k | 55.9k | 1.4k | 1.9k |
>
> We give the results (in thousands of US dollars) for 5 evenly-spaced points on the age profile in the table above. Whereas the naive estimate (without boosting) features a steep decline of \$11k from ages 65-89, our boosted estimates are substantially flatter --- better matching the theoretical model for pension and social security income from Kaplan and Violante. Furthermore, while the naive confidence intervals shrink for the highest ages, our boosted confidence intervals actually grow slightly, suggesting that the naive model may undercover for the oldest part of the age profile.
>
> [2] Kaplan, Greg, and Giovanni L. Violante. "A model of the consumption response to fiscal stimulus payments." Econometrica 82.4 (2014): 1199-1239.
>
> # Reviewer-Specific Response
>
> We appreciate your detailed feedback on our submission. We hope that the generalization from linear functions to more flexible functional forms addresses your concern over the class C. We provide responses to some of your specific comments below.
>
> Regarding Weaknesses 2 and 3, we agree that the proof techniques are not fundamentally new. One of our main goals in the paper is to connect the two very separate literatures on multiaccuracy/multicalibration and semiparametric efficiency.  We believe this has lead to some exciting and novel research directions.
>
> For Theorem 1, our goal was to establish that once-boosted ridge satisfies the multi-accuracy criterion. While the broader multi-accuracy literature often uses ridge boosting, it is usually performed with multiple boosting iterations and with an exponential weighting-type update. We wanted to verify that our simpler ridge boosting procedure satisfies the same multi-accuracy property. We will update the text to further clarify this point.
>
> For Theorem 2, the result indeed follows from applying Chernozhukov et al. (2023) together with Singh (2024): the novelty lies in recognizing that a single model (once-boosted ridge) satisfies the requirements of Chernozhukov et al. (2023) simultaneously for many different estimands. In particular, the debiased estimator outlined in Chernozhukov et al. (2023) requires estimating two nuisances: one for the outcome and one for the particular estimand of interest. It then guarantees semiparametric efficiency for that one estimand. Our Theorem then shows that once-boosted ridge provides semiparametric efficiency for *every* estimand in which the corresponding $dQ/dP$ or Riesz representer is in the Hilbert space ball $\mathcal{C}$.
>
> Finally, we appreciate you pointing out the issues in Weaknesses 4-6. We have focused on tightening notation in our revision. The probabilistic limit on $dQ/dP - c$ and $\Psi(Q)$ in Theorem 2 are both typos that we have fixed.

---

> > ### Comment · Reviewer_6txf · 2025-08-08
> > **Response to rebuttal**
> >
> > Thanks for the clear and thoughtful rebuttal. Expanding $\mathcal{C}$ from raw-linear to linear in rich features $\phi(X)$ (RKHS, sample-split adaptive bases, etc.) makes the setting feel much more realistic, and the “one model for many estimands” property is now better motivated. The new simulations and the ACS example help ground the contribution.
> >
> > Theorem 1 is still mainly a verification, and Theorem 2 still leans heavily on prior work, but I now see more value in the connection you’re drawing between multiaccuracy and semiparametric efficiency, especially with the broader scope. I’m fine with raising my score by one notch. I will also reduce confidence though. While I'm not opposed to accepting this paper, I just don't have a strong enough sense for it to argue significantly for its acceptance.

---

> > > ### Author Response · Authors · 2025-08-08
> > >
> > > We are glad that you find our generalized setting more realistic, making the connection between the two literatures more valuable. Thank you for the helpful feedback; we think the revised draft is stronger as a result.

---

> ### Author Response · Authors · 2025-08-05
>
> Since the discussion period has been extended, please let us know if there are any questions about our rebuttal that we could answer.

---

> ### Author Response · Authors · 2025-08-08
>
> Please let us know if you have remaining concerns or questions about our paper. We hope that your main concern about a linear model for dQ/dP has been addressed: our updated results outlined in the rebuttal also apply to boosting with high dimensional feature transformations, kernel ridge, and random forests. In our revised draft we have also focused on tightening up the technical presentation.

---

### Official Review · Reviewer_SMVd · 2025-07-05

**Clarity:** 3
**Significance:** 2
**Originality:** 3
**Rating:** 4
**Confidence:** 2

**Summary:**

This paper bridges fairness-inspired multiaccuracy estimation with semiparametric efficiency from causal inference. It shows that when the auditing class is a Hilbert space, a simple ridge-boosting step applied to the residuals of a base estimator yields a multiaccurate predictor that is numerically equivalent to an augmented balancing weight (ABW) estimator. As a result, this estimator inherits both universal adaptability (robustness under unknown covariate shifts) and semiparametric efficiency (minimum asymptotic variance), even without access to target distribution data. The authors provide theoretical justification, draw connections to Riesz regression and TMLE, and validate their claims through simulations demonstrating improved confidence interval coverage under distribution shift.

**Questions:**

Have you evaluated the proposed estimator on real datasets with covariate shift? If not, do you anticipate any challenges in doing so?

Your simulation assumes a linear-Gaussian data-generating process. How do you expect performance to change in more complex, nonlinear settings?

**Ethical Concerns:**

["NO or VERY MINOR ethics concerns only"]

**Final Justification:**

Thank you for the detailed response. The expanded theory, broader applicability, and practical demonstrations address my concerns. I am revising my score to borderline accept.

**Limitations:**

This is a theoretical contribution that adds conceptual clarity and unifies ideas across fairness, robustness, and causal inference. Its practicality could be enhanced by empirical validation on real datasets and extending beyond linear function classes.

**Quality:**

3

**Strengths And Weaknesses:**

Strengths:
The paper establishes a link between multiaccuracy and semiparametric efficiency, uniting fairness and causal inference methodologies.

The estimator does not require access to target distribution data, making it highly applicable in privacy-sensitive or resource-constrained settings.

The method is simple, which implements a practical, one-step ridge-boosting procedure that is easy to use and analyze.

At last, the paper provides empirical evidence that the once-boosted estimator achieves valid confidence interval coverage under covariate shifts.

Weakness:
Only one synthetic simulation setting is presented, with no real-world benchmark or application to support claims of practical effectiveness.

Key efficiency results rely on assumptions such as consistent outcome regression and the density ratio being well-approximated within the Hilbert space. In practice, these assumptions may not hold, and there is no empirical test or sensitivity analysis of robustness to assumption violations.

While the connection to existing debiasing methods is insightful, the novelty lies more in interpretation than in introducing a new estimator. The paper might be perceived as a reinterpretation of known techniques rather than a fundamentally new method.

---

> ### Author Rebuttal · Authors · 2025-07-30
>
> # General Response
>
> ## Expanded Results:
>
> We appreciate reviewers' thoughtful engagement with our submission. To streamline exposition in our initial submission, we characterized the results in terms of linear functions of $X$ and focused on the estimand $\mathbb{E}_Q[Y]$. In revisiting our technical results, we realized that we can substantially generalize both the class of functions and target estimands we consider --- with essentially no changes to the proofs at all. We believe that these revised results are responsive to reviewer questions and greatly expand the scope of our paper.
>
> * **Beyond linearity in $X$.** First, our results hold exchanging linear functions of $X$ for linear functions in features $\phi(X)$, which is a remarkably broad class of functions. This class includes when $\phi$ are the infinite-dimensional features from an RKHS; in this case the resulting algorithm would be once-boosted kernel ridge regression. This class also includes adaptive bases that are selected via sample splitting, such as honest random forests and ridge regression on the last-layer embedding of a pre-trained neural network.
>
>      An important limitation is that the results using our proof techniques are not fully general and do not hold for adaptive bases without sample splitting. That said, there are other examples where boosting also implicitly estimates a density ratio that do not rely on ridge regression or sample splitting. One prominent instance is random forests, which Lin and Han [1] show implicitly estimate a density ratio. In our revised manuscript, we adapt their results to show that, under certain restrictions, random forests satisfy the requirements in our Theorem 2; and therefore that boosting with random forests enjoys the same properties as boosting with ridge. While we leave a thorough discussion of this to future work, we think this is a very exciting direction that opens up many promising research avenues.
>
>
> * **More general estimands** Second, we show that our results continue to hold when we replace the estimand $\mathbb{E}_Q[Y]$ with any linear functional of $\mathbb{E}[Y|X]$, including average derivatives, average impulse responses, average treatment effects and more. To do so, we modify the class $\mathcal{C}$ to contain the Riesz representers for these functionals: a special case is the density ratio $dQ/dP$, which is the Riesz representer for $\mathbb{E}_Q[Y]$. Again, we achieve this generality without changing the proofs.
>
>      One important upshot of this generalization is that we can simultaneously consider a vector of estimands if each has a Riesz representer in the set $\mathcal{C}$. Many applied problem estimate a vector of estimands, and existing semiparametric estimators require fitting a separate debiasing model for each individual point in the vector. Our once-boosted ridge procedure only requires fitting a single model, which can then be used to construct the whole vector of estimates. This is practically very appealing, although a thorough assessment of this is beyond the scope of the current paper, especially with respect to valid joint inference. In particular, in follow up work we will better articulate conditions under which the 95\% confidence band over the whole vector is simultaneously valid.
>
> [1] Lin, Zhexiao, and Fang Han. "On regression-adjusted imputation estimators of the average treatment effect." arXiv preprint arXiv:2212.05424 (2023).
>
> ## Difficult Average Derivative Simulation:
>
> To demonstrate our results in a more general setting, we consider a non-linear simulation where we estimate the average derivative across multiple distributions (combining distribution shift and a new estimand). For these initial results, we retain a synthetic setup because we would like to show coverage of the confidence intervals. We choose a setting with 3 covariates: $X_1, X_2$ are iid $N(\mu,1)$, $X_3 = 4 \cdot \sigma(X_1 - X_2) + \epsilon - 2$, where $\sigma(\cdot)$ is the sigmoid, and $\epsilon \sim N(0,2)$. For the training distribution $\mu = 0$.
>
> The outcomes follow: $Y \sim f(X) + \eta$, where $f(X) = X_1 \cdot( 0.2 + \sin(X_1) + \sigma(X_2) - 0.2 \cdot X_3)$ and $\eta \sim N(0,2)$. The estimand is the average derivative under $Q$, $\mathbb{E}_Q[ \partial f(X) / \partial X_1 ]$. We consider three different distributions for $Q$. Each is identical to the setup above but changes the value of $\mu \in \{-1,0,1\}$. The dependence of $X_3$ on both $X_1$ and $X_2$ makes the average derivative very difficult to estimate, as we will show.
>
> We fit kernel ridge regression (with an RBF kernel) of $Y$ on $X$ in the training set. Then we also fit a once-boosted kernel ridge regression on the training set. For each we compute estimates of the average derivative of the model under the three Q distributions. The nominal (not boosted) krr estimate *badly* undercovers: with $n = 500$, the $95\\%$ coverage for the three target estimands is $49\\%$, $70\\%$, and $31\\%$ respectively. Our boosted estimator achieves coverage $93\\%$, $95\\%$, $92\\%$ respectively. In this challenging non-linear setting the gap in performance is actually much larger than in the linear ridge setting we originally submitted. (The final submission will include full figures for this experiment, which we cannot include in the rebuttal.)
>
> ## Real Data Experiment
>
> Following reviewer feedback, we also applied this approach to a real data example using American Community Survey (ACS) data, where the goal is to estimate counterfactual outcome means across a vector of ages. In this setting, $Y$ is income, $A$ is age, and $X$ are covariates like occupation, marriage status, and place of birth. Let $f(A,X) = \mathbb{E}[Y|A,X]$. Our estimands are $\mathbb{E}[f(65, X)], \mathbb{E}[f(66, X)], ... \mathbb{E}[f(90, X)]$. This vector of estimands represents the "age profile" of total income in retirement keeping $X$ fixed. This profile is an important input in many macroeconomic models (see, for example, the seminal paper from Kaplan and Violante [2]), especially for modeling consumption in retirement.
> Each of these estimands has substantial covariate shift when focusing on ages over 65.
> Typical modeling here would construct separate semiparametric efficient estimators for each point in the age profile, necessarily requiring a separate debiasing term at every age.
> We can instead use our theoretical results to motivate a common debiasing procedure for the entire vector of estimands: in our example we fit kernel ridge regression to estimate $f(A,X)$ and then use one round of boosting. Finally, we use the boosted $\hat{f}$ to compute a semiparametric-efficient estimate of every point on the age profile with corresponding confidence intervals.
>
> | Age | Naive Estimate | Boosted Estimate | Naive C.I. Width | Boosted C.I. Width |
> | --- | -------------- | ---------------- | ---------------- | ------------------ |
> | 65 | 61.1k | 59.5k | 1.7k | 1.8k |
> | 71 | 60.6k | 58.8k | 1.7k | 1.8k |
> | 77 | 58.5k | 57.6k | 1.6k | 1.8k |
> | 83 | 55.1k | 56.5k | 1.5k | 1.8k |
> | 89 | 50.8k | 55.9k | 1.4k | 1.9k |
>
> We give the results (in thousands of US dollars) for 5 evenly-spaced points on the age profile in the table above. Whereas the naive estimate (without boosting) features a steep decline of \$11k from ages 65-89, our boosted estimates are substantially flatter --- better matching the theoretical model for pension and social security income from Kaplan and Violante. Furthermore, while the naive confidence intervals shrink for the highest ages, our boosted confidence intervals actually grow slightly, suggesting that the naive model may undercover for the oldest part of the age profile.
>
> [2] Kaplan, Greg, and Giovanni L. Violante. "A model of the consumption response to fiscal stimulus payments." Econometrica 82.4 (2014): 1199-1239.
>
> # Reviewer-Specific Response
>
> Thank you for your feedback. We hope that the discussion above addresses some of your concerns about extending beyond linear function classes and demonstrating practical usage on a real dataset.
>
> We agree that this is not a fundamentally new method --- boosting with ridge regression is well-studied in the multicalibration literature. Our result instead establishes a new property for boosting with ridge regression: that it can be simultaneously semiparametrically efficient for all estimands described by the set $\mathcal{C}.$ This is in contrast to usual algorithms for semiparametric efficiency, which require estimating a different nuisance parameter specific to each estimand of interest. Previous work on multiaccurate estimators like ridge boosting have focused on robustness or fairness considerations, so it is surprising that the same procedure could achieve efficiency.
>
> Finally, we appreciate the question about a specification test or sensitivity analysis regarding violations of our assumptions. The core assumption required for our once-boosted ridge estimate to achieve Theorem 2 for a particular $Q$ is that $dQ/dP$ must belong to the Hilbert space ball $\mathcal{C}$. An infinite-dimensional Hilbert space can be extremely flexible, so this is not so much a question of model specification. However, it does crucially hinge on the radius of the Hilbert space ball. For a $Q$ with intense distribution shift, $dQ/dP$ could be very large, and so the radius must be very large. Practically speaking, this corresponds to making the regularization parameter for the boosting step very small. As a result, the variance of the estimate would grow substantially, resulting in a very large and even uninformative confidence interval, accurately reflecting the uncertainty of transferring to a very different $Q$. It would be useful to build a specification test/sensitivity analysis that allows us to check, how small we need to make the regularization parameter for the boosting step. Developing a feasible test for this condition would be an important topic for future work.

---

> ### Author Response · Authors · 2025-08-05
>
> Since the discussion period has been extended, please let us know if there are any questions about our rebuttal that we could answer.

---

### Official Review · Reviewer_cMtW · 2025-07-08

**Clarity:** 3
**Significance:** 4
**Originality:** 3
**Rating:** 4
**Confidence:** 2

**Summary:**

This paper connects two subareas of machine learning, multiaccuracy and semiparametric efficiency. It focuses on the case where the class of tests C (for multiaccuracy) is a Hilbert space. With some additional niceness assumptions, the ridge boosting estimator is multi-accurate and semiparametrically efficient. Together, these properties imply that the estimator has small bias and asymptotically efficient variance on unseen distributions Q (i.e., not used for training), for Q induced by the set of tests C.

This paper is in a line of very recent papers (last couple years) connecting multicalibration and causal inference. For example, Kim et al. 2022 showed that multiaccurate predictors can achieve low bias across certain (unseen) target populations, even when the distribution shift is unknown. However, multiaccuracy alone only guarantees bounds on the bias, not also the variance. In this paper, the Hilbert ball assumption is used to help bound the variance. Theorem 1 bounds the multi-accuracy error, and Theorem 2 bounds the bias and variance.

The paper also includes an experiment with synthetic data, comparing the coverage of single ridge outcome regression and ridge regression with auditing. On the synthetic data, with the same tuning parameter lambda, the boosted ridge has better coverage per sample size.

**Questions:**

---With respect to the assumptions 1-5 in the paper:
-Why is assumption 5 (supplemental pg 2) necessary?
-On pg 7, main body, assumption 3 and 4 are claimed to be standard in the semiparametric efficiency literature. What is an example of a problem for which some of the assumptions fail, and this argument cannot be applied? Are there "natural" extensions (e.g., slightly different problem settings) that are not captured by this approach (and perhaps worthy of future investigation)?


---What other auditing function classes (besides Hilbert balls) are plausible candidates for connecting multiaccuracy (or multicalibration definitions more generally) to the semiparametric efficiency literature?

---Minor typos:
Line 228: derive -> deriving
Line 319: are results -> our results
---How would this algorithm perform on a "real-world" data set? Which applications could plausibly satisfy (or be close to satisfying) the 5 assumptions necessary for Theorem 2? To which applications is the estimator in this paper particularly well-suited? (e.g., If you had to pick one "real-world" application, what would it be?)

**Ethical Concerns:**

["NO or VERY MINOR ethics concerns only"]

**Final Justification:**

Copying the following from my comment to the authors, after their initial rebuttal to my review:

Having read the other reviewers' evaluations, comments, and rebuttals, my overall evaluation of the paper has not significantly changed. I am positive towards this paper bridging two subareas (multiaccuracy and semiparametric efficiency) and leading to further interesting research.

While the high-level contributions seem impressive and above the bar for acceptance, admittedly, the technical content is difficult for me to follow. Therefore, it is hard for me to confidently suggest an evaluation for this submission. I am willing to defer to reviewers EDaD and wM69's positive reviews, trusting that their domain knowledge and understanding of this submission's merits greatly outstrips my own. Moreover, at the time of writing this comment, it appears that the negative comments of Reviewer 6txf have not been resolved, and I do not currently have a nuanced understanding of their disagreements with the authors.

Nevertheless, I do not feel personally comfortable increasing my confidence or rating of the submission

**Limitations:**

Yes.

**Quality:**

3

**Strengths And Weaknesses:**

Quality:
The claims are primarily theoretical and supported by proofs. The proofs appear to be complete and correct. The paper makes some claims about the practicality of multiaccurate learning algorithms (in general), but the experiment that appears in this work is only on synthetic data.

Clarity:
The paper is nicely organized, and the background sections set the stage well for the technical content of the paper. Additional context in the exposition of the scope of the work's results would be appreciated, as discussed by my review and other reviewers.

Significance:
This paper attempts to connect the multicalibration and semiparametrics literatures. In the special case when the multiaccuracy class is the Hilbert ball, ridge regression leads to an estimator that implicitly performs Riesz regression. It shows that the notions from multiaccuracy and semiparametric efficiency can coincide, and that there may be other possible connections between these two subareas. This could lead to future projects by researchers from either areas.

Originality:
Many comments about significance above also apply for originality. The paper draws heavily from both multicalibration and semiparametrics, as it should for a paper attempting to combine ideas from both areas. Relevant citations are provided to distinguish this work from previous work.

---

> ### Author Rebuttal · Authors · 2025-07-30
>
> # General Response
>
> ## Expanded Results:
>
> We appreciate reviewers' thoughtful engagement with our submission. To streamline exposition in our initial submission, we characterized the results in terms of linear functions of $X$ and focused on the estimand $\mathbb{E}_Q[Y]$. In revisiting our technical results, we realized that we can substantially generalize both the class of functions and target estimands we consider --- with essentially no changes to the proofs at all. We believe that these revised results are responsive to reviewer questions and greatly expand the scope of our paper.
>
> * **Beyond linearity in $X$.** First, our results hold exchanging linear functions of $X$ for linear functions in features $\phi(X)$, which is a remarkably broad class of functions. This class includes when $\phi$ are the infinite-dimensional features from an RKHS; in this case the resulting algorithm would be once-boosted kernel ridge regression. This class also includes adaptive bases that are selected via sample splitting, such as honest random forests and ridge regression on the last-layer embedding of a pre-trained neural network.
>
>      An important limitation is that the results using our proof techniques are not fully general and do not hold for adaptive bases without sample splitting. That said, there are other examples where boosting also implicitly estimates a density ratio that do not rely on ridge regression or sample splitting. One prominent instance is random forests, which Lin and Han [1] show implicitly estimate a density ratio. In our revised manuscript, we adapt their results to show that, under certain restrictions, random forests satisfy the requirements in our Theorem 2; and therefore that boosting with random forests enjoys the same properties as boosting with ridge. While we leave a thorough discussion of this to future work, we think this is a very exciting direction that opens up many promising research avenues.
>
> * **More general estimands** Second, we show that our results continue to hold when we replace the estimand $\mathbb{E}_Q[Y]$ with any linear functional of $\mathbb{E}[Y|X]$, including average derivatives, average impulse responses, average treatment effects and more. To do so, we modify the class $\mathcal{C}$ to contain the Riesz representers for these functionals: a special case is the density ratio $dQ/dP$, which is the Riesz representer for $\mathbb{E}_Q[Y]$. Again, this extra generality does not change the proofs.
>
>      One important upshot of this generalization is that we can simultaneously consider a vector of estimands if each has a Riesz representer in the set $\mathcal{C}$. Many applied problems estimate a vector of estimands, and existing semiparametric estimators require fitting a separate debiasing model for each element of the vector. Our once-boosted ridge procedure only requires fitting a single model, which can then be used to construct the whole vector of estimates. This is practically very appealing, although a thorough assessment of this is beyond the scope of the current paper, especially with respect to valid joint inference.
>
> [1] Lin, Zhexiao, and Fang Han. "On regression-adjusted imputation estimators of the average treatment effect." arXiv preprint arXiv:2212.05424 (2023).
>
> ## Difficult Average Derivative Simulation:
>
> To demonstrate our results in a more general setting, we consider a non-linear simulation where we estimate the average derivative across multiple distributions (combining distribution shift and a new estimand). For these initial results, we retain a synthetic setup because we would like to show coverage of the confidence intervals. We choose a setting with 3 covariates: $X_1, X_2$ are iid $N(\mu,1)$, $X_3 = 4 \cdot \sigma(X_1 - X_2) + \epsilon - 2$, where $\sigma(\cdot)$ is the sigmoid, and $\epsilon \sim N(0,2)$. For the training distribution $\mu = 0$.
>
> The outcomes follow: $Y \sim f(X) + \eta$, where $f(X) = X_1 \cdot( 0.2 + \sin(X_1) + \sigma(X_2) - 0.2 \cdot X_3)$ and $\eta \sim N(0,2)$. The estimand is the average derivative under $Q$, $\mathbb{E}_Q[ \partial f(X) / \partial X_1 ]$. We consider three different distributions for $Q$. Each is identical to the setup above but changes the value of $\mu \in \{-1,0,1\}$. The dependence of $X_3$ on both $X_1$ and $X_2$ makes the average derivative very difficult to estimate, as we will show.
>
> We fit kernel ridge regression (with an RBF kernel) of $Y$ on $X$ in the training set. Then we also fit a once-boosted kernel ridge regression on the training set. For each we compute estimates of the average derivative of the model under the three Q distributions. The nominal (not boosted) krr estimate *badly* undercovers: with $n = 500$, the $95\\%$ coverage for the three target estimands is $49\\%$, $70\\%$, and $31\\%$ respectively. Our boosted estimator achieves coverage $93\\%$, $95\\%$, $92\\%$ respectively. In this challenging non-linear setting the gap in performance is actually much larger than in the linear ridge setting we originally submitted. (The final submission will include full figures for this experiment, which we cannot include in the rebuttal.)
>
> ## Real Data Experiment
>
> Following reviewer feedback, we also applied this approach to a real data example using American Community Survey (ACS) data, where the goal is to estimate counterfactual outcome means across a vector of ages. In this setting, $Y$ is income, $A$ is age, and $X$ are covariates like occupation, marriage status, and place of birth. Let $f(A,X) = \mathbb{E}[Y|A,X]$. Our estimands are $\mathbb{E}[f(65, X)], \mathbb{E}[f(66, X)], ... \mathbb{E}[f(90, X)]$. This vector of estimands represents the "age profile" of total income in retirement keeping $X$ fixed. This profile is an important input in many macroeconomic models (see, for example, the seminal paper from Kaplan and Violante [2]), especially for modeling consumption in retirement.
> Each of these estimands has substantial covariate shift when focusing on ages over 65.
> Typical semiparametric estimators would require constructing a separate debiasing term at every age. We can instead use our theoretical results to motivate a common debiasing procedure for the entire vector of estimands: in our example we fit kernel ridge regression to estimate $f(A,X)$ and then use one round of boosting. Finally, we use the boosted $\hat{f}$ to compute a semiparametric-efficient estimate of every point on the age profile with corresponding confidence intervals.
>
> | Age | Naive Estimate | Boosted Estimate | Naive C.I. Width | Boosted C.I. Width |
> | --- | ----- | ----- | ----- | ---- |
> | 65 | 61.1k | 59.5k | 1.7k | 1.8k |
> | 71 | 60.6k | 58.8k | 1.7k | 1.8k |
> | 77 | 58.5k | 57.6k | 1.6k | 1.8k |
> | 83 | 55.1k | 56.5k | 1.5k | 1.8k |
> | 89 | 50.8k | 55.9k | 1.4k | 1.9k |
>
> We give the results (in thousands of US dollars) for 5 evenly-spaced points on the age profile in the table above. Whereas the naive estimate (without boosting) features a steep decline of \$11k from ages 65-89, our boosted estimates are substantially flatter --- better matching the theoretical model for pension and social security income from Kaplan and Violante. Furthermore, while the naive confidence intervals shrink for the highest ages, our boosted confidence intervals actually grow slightly, suggesting that the naive model may undercover for the oldest part of the age profile.
>
> [2] Kaplan, Greg, and Giovanni L. Violante. "A model of the consumption response to fiscal stimulus payments." Econometrica 82.4 (2014): 1199-1239.
>
> # Reviewer-Specific Response
>
> We are grateful for your thoughtful engagement with our paper and for the constructive feedback.
>
> **Assumptions.**  First, we agree that our initial submission did not provide sufficient intuition for the underlying assumptions. We give some additional intuition here, and will include longer responses in the updated manuscript.
>
> Assumption 5 essentially requires that the higher-order moments of the efficient influence function are bounded. This is a minimal condition on the higher-order moments of the efficient influence function such that the variance $\hat{V}$ asymptotically converges to the true variance and therefore provides a valid confidence interval. Most central limit theorem-based confidence intervals require a similar condition. This condition is mostly like to be violated in heavy-tailed data.
>
> Assumption 3 bounds variances for the outcome. These conditions are very likely to be satisfied unless the $X$ distribution or conditional noise distribution have very heavy tails (like a Cauchy distribution). Assumption 4 is also a relatively weak requirement. This assumption merely states that given an infinite amount of data, the estimate of the density ratio will eventually converge to the truth (and that this can happen at an arbitrarily slow rate). This assumption is very plausible using the flexible basis functions that we describe in the general response above.
>
> Overall, Assumptions 3-5 will be satisfied in most empirical applications except when dealing with heavy tailed data (such as on some stock return data sets, weather data with extreme events, etc).
>
> The key substantive assumptions are unconfoundedness and overlap, and these do not hold for every application. These common observational causal inference assumptions are more likely to hold (approximately) in settings like electronic medical records with rich covariate information and enough overlap in the $X$ distributions from varied treatment assignment. In terms of other applications, the ``age profile'' example that we detailed above is inspired, although in extremely simplified form, by a real empirical application (currently unpublished ongoing work) where fitting a debiasing nuisance for each point on the age profile would be computationally-intensive and add additional variance. Using this boosting procedure in that setting would be quite helpful.
>
> Finally: thank you for catching those typos!

---

> > ### Comment · Reviewer_cMtW · 2025-08-08
> >
> > Thanks to the authors for this rebuttal. The clarification of the assumptions was helpful for me in better understanding the scope of this work. As the authors said, I hope this discussion and similar clarifications of the scope of the work make it to an updated version of the manuscript.
> >
> > Having read the other reviewers' evaluations, comments, and rebuttals, my overall evaluation of the paper has not significantly changed. I am positive towards this paper bridging two subareas (multiaccuracy and semiparametric efficiency) and leading to further interesting research.
> >
> > While the high-level contributions seem impressive and above the bar for acceptance, admittedly, the technical content is difficult for me to follow. Therefore, it is hard for me to confidently suggest an evaluation for this submission. I am willing to defer to reviewers EDaD and wM69's positive reviews, trusting that their domain knowledge and understanding of this submission's merits greatly outstrips my own. Moreover, at the time of writing this comment, it appears that the negative comments of Reviewer 6txf have not been resolved, and I do not currently have a nuanced understanding of their disagreements with the authors.
> >
> > Nevertheless, I do not feel personally comfortable increasing my confidence or rating of the submission. Thanks again to the authors for their engagement during the rebuttal period.

---

> > > ### Author Response · Authors · 2025-08-08
> > >
> > > We are glad to hear that you find the high-level contributions impressive, and we agree that we are particularly excited to see if this paper will lead to further research between the two fields.
> > >
> > > It looks like the negative comments of Reviewer 6txf were resolved by our rebuttal, especially the fact that we use rich non-linear features in our new average derivative task and ACS income example. We hope that assuages any additional concerns you might have.
> > >
> > > Thank you again for the discussion.

---

> ### Author Response · Authors · 2025-08-05
>
> Since the discussion period has been extended, please let us know if there are any questions about our rebuttal that we could answer.

---

### Official Review · Reviewer_EDaD · 2025-07-24

**Clarity:** 2
**Significance:** 2
**Originality:** 2
**Rating:** 5
**Confidence:** 2

**Summary:**

In this paper, the authors analyze properties of "multiaccurate estimators" in the covariate shift setting (labeled source data, unlabeled target data, the distribution of features $X$ changes). With the goal of estimating the expectation of Y (label) in the target distribution, and for a defined "auditing function class" (in their case a Hilbert ball), the authors analyze a previously derived multiaccurate model using a ridge boosting estimator (an additive term that corrects a model fit to the labeled source data using, e.g., ERM). The authors show the ridge boosted multiaccurate estimator is semiparametrically efficient for any member of the Hilbert ball (and equivalently, any corresponding covariate shift-correcting density ratio). This means that despite needing to estimate the source outcome regression function (\gamma), and the auditing functions, the boosted multiaccurate estimator of $E_Q[Y]$ is asymptotically $\sqrt{N}$-consistent and achieves the optimal variance bound. In a simple linear regression simulation, the authors show that the boosted ridge regression estimator achieves correct confidence interval coverage as opposed to undercoverage by standard ridge regression.

**Questions:**

Q1: If, as the authors state, one lets the auditing function class be a "collection of density ratios", how does multiaccuracy differ from Distributionally robust optimization (DRO) (e.g., [1, 2]) (Equations 3 and 4 in [1] might be most relevant.)? In DRO we might be interested in minimizing the maximum error across distributions in a bounded density-ratio ball. If this worst-case error is below $\alpha$, and the density ratio ball is the same as the auditing function class C, is the solution to this DRO problem a multiaccurate estimator? If so, discussion and comparison to DRO approaches seems interesting and relevant. If they are related, how does your efficiency result compare to the statistical properties of DRO estimators under covariate shift?


[1] Duchi, J., Hashimoto, T., & Namkoong, H. (2023). Distributionally robust losses for latent covariate mixtures. Operations Research, 71(2), 649-664.

[2] Duchi, J. C., & Namkoong, H. (2021). Learning models with uniform performance via distributionally robust optimization. The Annals of Statistics, 49(3), 1378-1406.

**Ethical Concerns:**

["NO or VERY MINOR ethics concerns only"]

**Final Justification:**

The authors addressed my concerns (not readable without Kim et al (2022); engaged in discussion about DRO) in their rebuttal. As such I have raised my score.

**Limitations:**

Yes

**Quality:**

3

**Strengths And Weaknesses:**

For context, I reviewed this as someone who has done a lot of work on model training and evaluation under distribution shifts, and who is familiar with the causal inference literature. Notably, I did not have prior familiarity with multiaccuracy (though I had heard of multicalibration).

### Clarity:
- I think the biggest weakness of the paper is that it is written in a way where it heavily depends on the reader being intimately familiar with multiaccuracy literature. This may be the simplest presentation of the material, since it demonstrates a property of the ridge boosting estimator from Kim et al. (2022), but this means that the paper does not stand well on its own without having read the original paper first.

### Originality/Significance:
- Because the paper is written to heavily rely on familiarity with Kim et al. (2022), the potential significance and impact of this paper is much lower than it might otherwise be. To grossly oversimplify, it seems the main result is to show that, for a particular class of auditing functions, the estimator from Kim et al. (2022) achieves semiparametric efficiency "for free." I will let a more theoretically knowledgeable reviewer speak to the significance of this result, but speaking from the perspective of a reader interested in theory + applications of distribution shift who is not as well-steeped in this literature, the paper does not speak to why this result is surprising, or what this result makes possible that could not be done before.
- I think the authors could better motivate the choice of $E_Q[Y]$ as their estimand, since $E_Q[Y]$ is not usually the quantity of interest in machine learning. That said, I can think of examples where it is useful. For example, estimating performance on new data (under covariate shift) in settings where labels can't be easily obtained in a timely manner (such as post-deployment monitoring for long term risk predictors). Or estimating the average treatment effect under a covariate shift.
- The form of multiaccuracy in Definition 1, and its relation to a "collection of density ratios" (Ln 144) makes it look quite similar to the form of Distributionally Robust Optimization. For readers familiar with DRO but not with multiaccuracy, it would be useful for the authors to include a discussion of the similarities and differences.

### Quality:
- I don't have much to comment on re: quality. I leave the more technical details to other reviewers.
- The simulated experiment confirms the asymptotic normality result of Theorem 2 in finite samples for a very simple linear regression setup.

---

> ### Author Rebuttal · Authors · 2025-07-30
>
> # General Response
>
> ## Expanded Results:
>
> We appreciate reviewers' thoughtful engagement with our submission. To streamline exposition in our initial submission, we characterized the results in terms of linear functions of $X$ and focused on the estimand $\mathbb{E}_Q[Y]$. In revisiting our technical results, we realized that we can substantially generalize both the class of functions and target estimands we consider --- with essentially no changes to the proofs at all. We believe that these revised results are responsive to reviewer questions and greatly expand the scope of our paper.
>
> * **Beyond linearity in $X$.** First, our results hold exchanging linear functions of $X$ for linear functions in features $\phi(X)$, which is a remarkably broad class of functions. This class includes when $\phi$ are the infinite-dimensional features from an RKHS; in this case the resulting algorithm would be once-boosted kernel ridge regression. This class also includes adaptive bases selected via sample splitting, such as honest random forests or last-layer embedding of a pre-trained neural network.
>
>      An important limitation is that the results using our proof techniques are not fully general and do not hold for adaptive bases without sample splitting. That said, there are other examples where boosting also implicitly estimates a density ratio that do not rely on ridge regression or sample splitting. One prominent instance is random forests, which Lin and Han [1] show implicitly estimate a density ratio. In our revised manuscript, we adapt their results to show that, under certain restrictions, random forests satisfy the requirements in our Theorem 2; and therefore that boosting with random forests enjoys the same properties as boosting with ridge. We think this is a very exciting direction that opens up many promising research avenues.
>
> * **More general estimands** Second, we show that our results continue to hold when we replace the estimand $\mathbb{E}_Q[Y]$ with any linear functional of $\mathbb{E}[Y|X]$, including average derivatives, average impulse responses, average treatment effects and more. To do so, we modify the class $\mathcal{C}$ to contain the Riesz representers for these functionals: a special case is the density ratio $dQ/dP$, which is the Riesz representer for $\mathbb{E}_Q[Y]$. Again, this extra generality does not change the proofs.
>
>      One important upshot of this generalization is that we can simultaneously consider a vector of estimands if each has a Riesz representer in the set $\mathcal{C}$. Many applied problems estimate a vector of estimands, and existing semiparametric estimators require fitting a separate debiasing model for each element of the vector. Our once-boosted ridge procedure only requires fitting a single model, which can then be used to construct the whole vector of estimates. This is practically very appealing, although in future work we would need to secure valid joint inference.
>
> [1] Lin, Zhexiao, and Fang Han. "On regression-adjusted imputation estimators of the average treatment effect." arXiv preprint arXiv:2212.05424 (2023).
>
> ## Difficult Average Derivative Simulation:
>
> To demonstrate our results in a more general setting, we consider a non-linear simulation where we estimate the average derivative across multiple distributions (combining distribution shift and a new estimand). We retain a synthetic setup because we would like to show coverage of the confidence intervals. We choose a setting with 3 covariates: $X_1, X_2$ are iid $N(\mu,1)$, $X_3 = 4 \cdot \sigma(X_1 - X_2) + \epsilon - 2$, where $\sigma(\cdot)$ is the sigmoid, and $\epsilon \sim N(0,2)$. For the training distribution $\mu = 0$.
>
> The outcomes follow: $Y \sim f(X) + \eta$, where $f(X) = X_1 \cdot( 0.2 + \sin(X_1) + \sigma(X_2) - 0.2 \cdot X_3)$ and $\eta \sim N(0,2)$. The estimand is the average derivative under $Q$, $\mathbb{E}_Q[ \partial f(X) / \partial X_1 ]$. We consider three different distributions for $Q$. Each is identical to the setup above but changes the value of $\mu \in \{-1,0,1\}$. The dependence of $X_3$ on both $X_1$ and $X_2$ makes the average derivative very difficult to estimate.
>
> We fit kernel ridge regression (with an RBF kernel) of $Y$ on $X$ in the training set plus one boosting step with kernel ridge. For each we compute estimates of the average derivative of the model under the three Q distributions. The nominal (not boosted) krr estimate *badly* undercovers: with $n = 500$, the $95\\%$ coverage for the three target estimands is $49\\%$, $70\\%$, and $31\\%$ respectively. Our boosted estimator achieves coverage $93\\%$, $95\\%$, $92\\%$ respectively. The improvement for the boosted estimator is actually larger than in the linear setting. (The final submission will include full figures for this experiment, which we cannot include in the rebuttal.)
>
> ## Real Data Experiment
>
> Following reviewer feedback, we also applied this approach to a real data example using American Community Survey (ACS) data, where the goal is to estimate counterfactual outcome means across a vector of ages. In this setting, $Y$ is income, $A$ is age, and $X$ are covariates like occupation, marriage status, and place of birth. Let $f(A,X) = \mathbb{E}[Y|A,X]$. Our estimands are $\mathbb{E}[f(65, X)], \mathbb{E}[f(66, X)], ... \mathbb{E}[f(90, X)]$. This vector of estimands represents the "age profile" of total income in retirement keeping $X$ fixed. This profile is an important input in many macroeconomic models (see, for example, the seminal paper from Kaplan and Violante [2]), especially for modeling consumption in retirement.
> Each of these estimands has substantial covariate shift when focusing on ages over 65.
> Typical semiparametric estimators would require constructing a separate debiasing term at every age. We can instead use our theoretical results to motivate a common debiasing procedure for the entire vector of estimands: in our example we fit kernel ridge regression to estimate $f(A,X)$ and then use one round of boosting. Finally, we use the boosted $\hat{f}$ to compute a semiparametric-efficient estimate of every point on the age profile with corresponding confidence intervals.
>
> | Age | Naive Estimate | Boosted Estimate | Naive C.I. Width | Boosted C.I. Width |
> | --- | --- | -- | -- | ---- |
> | 65 | 61.1k | 59.5k | 1.7k | 1.8k |
> | 71 | 60.6k | 58.8k | 1.7k | 1.8k |
> | 77 | 58.5k | 57.6k | 1.6k | 1.8k |
> | 83 | 55.1k | 56.5k | 1.5k | 1.8k |
> | 89 | 50.8k | 55.9k | 1.4k | 1.9k |
>
> We give the results (in thousands of US dollars) for 5 evenly-spaced points on the age profile in the table above. Whereas the naive estimate (without boosting) features a steep decline of \$11k from ages 65-89, our boosted estimates are substantially flatter --- better matching the theoretical model for pension and social security income from Kaplan and Violante. Furthermore, while the naive confidence intervals shrink for the highest ages, our boosted confidence intervals actually grow slightly, suggesting that the naive model may undercover for the oldest part of the age profile.
>
> [2] Kaplan, Greg, and Giovanni L. Violante. "A model of the consumption response to fiscal stimulus payments." Econometrica 82.4 (2014): 1199-1239.
>
> # Reviewer-Specific Response
>
> We appreciate the reviewer's push to engage more critically with DRO and to make this work more accessible to a broader audience.
>
> First, we agree that the exposition in the original submission relied too heavily on Kim et al. (2022). In the revision, we have expanded the background and setup so that readers unfamiliar with the earlier multiaccuracy results can nonetheless engage with the current work.
>
> **Question 1**: We thank the reviewer for this insightful question. Indeed, there is a close connection between **multiaccuracy** and **distributionally robust optimization (DRO)**, as also discussed in [3]. To clarify this connection, we first recall the standard DRO objective:
> $$
> \min_{f \in \mathcal{F}} \sup_{w \in \mathcal{C}} \mathbb{E}_P[w(X) \cdot \ell(f(X), Y)],
> $$
> where $w(X) = \frac{dQ}{dP}(X)$, $\mathcal{C}$ is a bounded density-ratio ball, and $\ell(f(X), Y)$ is the loss function of interest.
>
> In the special case where $\ell(f(X), Y) = Y - f(X)$, and the worst-case error is bounded by $\alpha$, as you noted, the solution to the DRO problem satisfies the multiaccuracy condition with respect to the function class $\mathcal{C}$ and the distribution $P$.
>
> Under the squared loss $\ell(f(X), Y) = (f(X) - Y)^2$, there is also a connection between multi-accuracy and the first-order optimality condition for DRO. Let $w^* \in \mathcal{C}$ denote the adversarial density ratio that attains the supremum. If $\mathcal{F}$ is a Hilbert space (e.g., an RKHS), the first-order optimality condition for the DRO objective becomes:
>
> $\mathbb{E}_P[w^*(X) \cdot (f(X) - Y) \cdot h(X)] = 0, \quad \forall h \in \mathcal{F}.$
>
> This is equivalent to the multiaccuracy condition under a reweighted distribution $\tilde{P}$, where $d\tilde{P}/dP = w^*$, i.e.,
> $$
> \mathbb{E}_{\tilde{P}}[(f(X) - Y) \cdot h(X)] = 0, \quad \forall h \in \mathcal{F}.
> $$
>
> Hence, under these assumptions, the DRO solution satisfies multiaccuracy with respect to the class $\mathcal{F}$ and the reweighted distribution $\tilde{P}$.
>
> Regarding efficiency, while many estimators can satisfy multiaccuracy, our proposed estimator is just one example. The DRO solution may take a different form. Our theoretical efficiency guarantees, which are specific to our construction, do not directly apply to general DRO estimators. Nonetheless, we agree that this connection is highly interesting and provides a promising direction for further exploration.
>
> We appreciate the reviewer highlighting this connection, and we will incorporate a discussion into the revised manuscript.
>
> [3] Hastings, J., Jung, C., Peale, C., $\&$ Syrgkanis, V. (2024). Taking a moment for distributional robustness. arXiv preprint arXiv:2405.05461.

---

> ### Author Response · Authors · 2025-08-05
>
> Since the discussion period has been extended, please let us know if there are any questions about our rebuttal that we could answer.

---

> > ### Comment · Reviewer_EDaD · 2025-08-06
> >
> > Thanks for your response. I think my primary concern will be addressed if in your revisions you significantly improve the clarity of the paper such that it is not so critically dependent on having read Kim et al. (2022). As a result I am raising my score.

---

> > > ### Author Response · Authors · 2025-08-06
> > >
> > > We have focused on rewriting the introduction to include more general motivation on distributional robustness that helps situate multiaccuracy/multicalibration papers within the larger literature that would be more accessible to people outside the fairness community. Thank you again.

---

### Note · Authors · 2025-08-13

Dear AC and reviewers,

We would like to summarize the main points of discussion regarding our paper.

Reviewers found our paper to be a valuable bridge between the multicalibration literature (in the fairness community) and the semiparametrics literature in causal inference and statistics. However, there were two repeated concerns across reviews.

First, regarding whether we require a linear functional form. In our rebuttal, we emphasize that the linear functional form was originally adopted for simplicity. In our revision, we show our result also applies to any high dimensional basis expansion, kernel ridge regression, random forests, and adaptive bases with sample splitting.

Second, multiple reviewers asked for further empirical validation. In the rebuttal, we discuss results on a new challenging non-linear benchmark and demonstrate that our performance improvements are larger in this setting than in the linear case. We also provide an empirical case study using ACS survey data.

We believe these revisions directly address the reviewers' concerns, broaden the paper's applicability, and provide stronger empirical and theoretical support for our contributions.

Thank you, all, for the valuable discussion.

---

### Decision · Program_Chairs · 2025-09-17

**Decision:**

Accept (spotlight)

**Comment:**

The authors illustrate how boosting estimators for improving multiaccuracy can be viewed as semiparametric efficient estimators of the subpopulation means across an auditing class, under the assumption that the auditing class is a Hilbert space. Reviewers found the connection established in this work between multiaccuracy and semiparametric theory/double machine learning to be insightful. As a reviewer mentioned, the connection can be obvious after the fact, but far from obvious initially.

I congratulate the authors on this interesting work and recommend acceptance of its work. I urge the authors to take into account the feedback from reviewers, particularly the suggestion to include additional empirical evaluations using real-world datasets.